# Rapid cell type-specific nascent proteome labeling in *Drosophila*

Stefanny Villalobos-Cantor[1], Ruth M Barrett[1], Alec F Condon[1†],
Alicia Arreola-Bustos[1], Kelsie M Rodriguez[2], Michael S Cohen[2], Ian Martin[1,2,3]*

[1]Jungers Center for Neurosciences, Department of Neurology, Oregon Health and Science University, Portland, United States; [2]Department of Chemical Physiology and Biochemistry, Oregon Health and Science University, Portland, United States; [3]Parkinson Center of Oregon, Oregon Health and Science University, Portland, United States

**\*For correspondence:**
martiia@ohsu.edu

**Present address:** †Department of Biology, Stanford University, Stanford, United States

**Competing interest:** The authors declare that no competing interests exist.

**Abstract** Controlled protein synthesis is required to regulate gene expression and is often carried out in a cell type-specific manner. Protein synthesis is commonly measured by labeling the nascent proteome with amino acid analogs or isotope-containing amino acids. These methods have been difficult to implement in vivo as they require lengthy amino acid replacement procedures. O-propargyl-puromycin (OPP) is a puromycin analog that incorporates into nascent polypeptide chains. Through its terminal alkyne, OPP can be conjugated to a fluorophore-azide for directly visualizing nascent protein synthesis, or to a biotin-azide for capture and identification of newly-synthesized proteins. To achieve cell type-specific OPP incorporation, we developed phenylace-tyl-OPP (PhAc-OPP), a puromycin analog harboring an enzyme-labile blocking group that can be removed by penicillin G acylase (PGA). Here, we show that cell type-specific PGA expression in *Drosophila* can be used to achieve OPP labeling of newly-synthesized proteins in targeted cell populations within the brain. Following a brief 2 hr incubation of intact brains with PhAc-OPP, we observe robust imaging and affinity purification of OPP-labeled nascent proteins in PGA-targeted cell populations. We apply this method to show a pronounced age-related decline in neuronal protein synthesis in the fly brain, demonstrating the capability of PhAc-OPP to quantitatively capture in vivo protein synthesis states. This method, which we call POPPi (*PGA-dependent OPP incorporation*), should be applicable for rapidly visualizing protein synthesis and identifying nascent proteins synthesized under diverse physiological and pathological conditions with cellular specificity in vivo.

## Editor's evaluation

The authors developed a versatile labeling strategy to allow visualization and identification of newly synthesized proteins in a cell population of interest. The approach enables cell-specific nascent proteome labeling from brain tissues to examine the role of translational control in different physiological and pathological states.

## Introduction

Controlled protein synthesis plays a fundamental role in orchestrating gene expression, and cellular protein amounts are thought to be primarily determined at the level of translation (*Schwanhäusser et al., 2011*). Translational control is integral to cellular development and homeostasis and becomes dysregulated in numerous disease states including cancer (*Robichaud and Sonenberg, 2017*), autism (*Gkogkas et al., 2013*), and neurodegeneration (*Wiebe et al., 2020*). Within a complex organ like the brain, the specialized and distinct properties of neuronal and glial subtypes arise from variable

expression of many protein components which are subject to translational control (*Doyle et al., 2008*). Regulation of global protein synthesis as well as the modulated synthesis of specific proteins at the synapse is important in synaptic plasticity and memory formation (*Sossin and Costa-Mattioli, 2019*).

Despite advances in assessing the transcriptome with cellular resolution, mRNA levels are an imprecise surrogate for protein abundance, and methods to capture the nascent proteome of individual cell populations in vivo have lagged behind. Current methods employ indiscriminate protein synthesis labeling in all cells of a tissue followed by flow sorting of tagged cell suspensions from these tissues (*Hidalgo San Jose and Signer, 2019*) or regional microdissection (*Griesser et al., 2020*) and suffer from a lack of precision or substantial loss of protein. An alternative method for quantifying protein synthesis is through non-canonical amino acid (NCAA) protein labeling, using methionine analogs. This approach recently progressed toward cell-type specificity through the expression of engineered methionyl-tRNA synthetase (MetRS) (*Alvarez-Castelao et al., 2017*; *Erdmann et al., 2015*; *Tanrikulu et al., 2009*). Only mutant MetRS can charge methionyl-tRNA with azidonorleucine (ANL) which is amenable to protein capture by BONCAT (biorthogonal non-canonical amino acid tagging) and visualization by FUNCAT (fluorescent non-canonical amino acid tagging) (*Alvarez-Castelao et al., 2017*; *Erdmann et al., 2015*; *Tanrikulu et al., 2009*). Hence, nascent protein labeling is restricted to cells that express mutant MetRS. However, this labeling strategy requires extended dietary methionine depletion and lengthy ANL feeding in mice or 1–2d of ANL feeding in flies, and chronic feeding is associated with significant developmental toxicity and behavioral deficits (*Alvarez-Castelao et al., 2017*; *Erdmann et al., 2015*).

A potentially more efficient approach to protein synthesis labeling is through the use of puromycin analogs (*Barrett et al., 2016*; *Liu et al., 2012*). Puromycin is structurally similar to tyrosyl tRNA, yet its incorporation is not amino acid-specific, hence in contrast to NCAA, its incorporation into nascent proteins is not biased by their sequence or extent of methionine content. This facilitates more uniform incorporation into newly-synthesized proteins (*Nathans, 1964*), which is advantageous in the assessment of global protein synthesis. One caveat to this approach is that puromycin incorporation results in premature chain termination and therefore the production of truncated puromycylated proteins that, while still generally amenable to MS-based proteomic analysis, may be targeted for proteasome-mediated degradation. Addition of an O-propargyl group to puromycin allows visualization or capture of newly-synthesized protein via click-chemistry conjugation to a fluorophore-azide or a biotin-tagged azide, respectively, and has been successfully demonstrated in cultured cells (*Liu et al., 2012*) and in cells isolated from OP-puromycin (OPP)-injected animals (*Hidalgo San Jose and Signer, 2019*). We developed an analog of OPP called PhAc-OPP that harbors an enzyme-labile blocking group (*Barrett et al., 2016*). This blocking group renders OPP incapable of nascent protein incorporation until its removal by the *E. coli* enzyme PGA (*Barrett et al., 2016*). Hence, targeted expression of PGA can, in principle, be used to limit OPP proteome labeling to individual cell populations within animal tissues. To directly test this, we generated PGA-transgenic *Drosophila* and developed a method that we call POPPi for rapid, cell type-specific labeling of protein synthesis in intact fly brains. Here, we show that POPPi is a versatile labeling strategy that can achieve efficient visualization and identification of newly-synthesized proteins in a cell population of interest. Our approach enables cell-specific nascent proteome labeling from complex brain tissue within just a few hours, making it a powerful and efficient tool for examining the role of translational control in different physiological and pathological states.

## Results

We previously demonstrated that neuronal PGA expression successfully converts PhAc-OPP to OPP, thus enabling OPP labeling of nascent proteins in cultured primary mouse neurons (*Barrett et al., 2016*; *Figure 1—figure supplement 1A*). Toward cell type-specific protein labeling in complex nervous system tissue in vivo, we generated a transgenic PGA fly line that can express FLAG-tagged PGA when crossed to any of the commonly available fly lines expressing a cell type-specific GAL4 driver of choice (*Figure 1—figure supplement 1B*). Following ubiquitous PGA expression, we assessed PGA levels in embryos and the brains of larvae, pupae, and adults. PGA expression is detectable in both developing and adult flies, with the highest expression levels seen in larval and pupal brains, followed by adult brains and embryos (*Figure 1—figure supplement 1C*). We reasoned that cell type-specific nascent protein labeling in the CNS might be efficiently achieved using intact *Drosophila* brain explants. *Drosophila* whole brains isolated and maintained ex vivo are remarkably

stable, exhibiting sustained neuronal morphology and physiological properties for many hours (*Ayaz et al., 2008*; *Brown et al., 2006*; *Essers et al., 2016*; *Gibbs and Truman, 1998*; *Gu and O'Dowd, 2006*; *Wang et al., 2003*; *Li et al., 2020*; *Lee et al., 2006*). Fly brain explants have been used to study neuronal activity such as response to odor stimulation (*Gu and O'Dowd, 2006*; *Wang et al., 2003*), axon remodeling (*Brown et al., 2006*; *Gibbs and Truman, 1998*), neuronal wiring (*Li et al., 2020*), neural stem cell proliferation (*Lee et al., 2006*), and protein synthesis (*Essers et al., 2016*). To examine whether isolated adult fly brains exhibit stable protein synthesis, we generated fly brain preparations and assessed $^{35}$S-methionine/cysteine incorporation levels over 8 hr. We observe no significant change in global protein synthesis rates over this time period (*Figure 1A*), suggesting that protein synthesis is stable in newly-isolated whole brain preparations and that these may therefore be suitable for capturing in vivo protein synthesis states.

## *Drosophila* brain nascent proteome labeling with OPP

Two major goals for cell-specific protein synthesis labeling are the visualization and identification of nascent proteomes in a cell type of interest. We hypothesized that this could be accomplished in *Drosophila* brain through OPP labeling of nascent polypeptide chains (NPC) followed by click-chemistry conjugation to either a fluorophore-azide for protein visualization or a biotin-azide for protein capture (schematic in *Figure 1B*). We first assessed whether unblocked OPP can label newly-synthesized proteins in *Drosophila* brain explants. Widespread OPP incorporation into newly-synthesized protein is clearly visible across the adult brain when coupled to the fluorophore-azide AF488-azide for detection (*Figure 1C*). The extent of incorporation is dependent on OPP concentration and slightly on incubation time (*Figure 1—figure supplement 2*) and is blocked by the protein synthesis inhibitor cycloheximide (*Figure 1C and D*), indicating that it is protein synthesis-dependent. Labeling is most prominently seen in the cell cortex, the location of cell bodies within the fly brain, consistent with the majority of protein synthesis occurring within the cell soma (*Figure 1C*). Theoretically, the protein synthesis imaged through this approach could represent OPP incorporation into a wide array of translating proteins or be restricted to a few highly-expressed proteins. In-gel fluorescence assessment of electrophoretically-separated brain extracts reveals that OPP-labeled proteins span the whole molecular weight range, while a minor degree of non-specific background AF488-azide incorporation is seen in the absence of OPP (*Figure 1E*). This finding suggests that OPP is efficiently and unbiasedly incorporated into a broad set of proteins, which is also consistent with prior studies from OPP labeling of cultured human cells (*Forester et al., 2018*).

## Rapid cell type-specific protein synthesis labeling with PhAc-OPP

We next assessed if targeted PGA expression can promote PhAc-OPP unblocking and OPP labeling of newly-synthesized proteins in cell populations of interest. We expressed PGA pan-neuronally in *Drosophila* brain using the *elavC155-GAL4* driver, which we confirmed by immunoblot (*Figure 2—figure supplement 1A*) and then treated freshly-isolated brain explants with PhAc-OPP for 2 hr. Robust neuronal protein synthesis labeling is seen following PhAc-OPP treatment, but not in the absence of PhAc-OPP (*Figure 2A*) or in PhAc-OPP treated brains in the absence of GAL4-driven PGA expression (*Figure 2—figure supplement 1B*), suggesting that PhAc-OPP is not substantially uncaged in the absence of PGA expression and therefore that it can be reliably used to label newly-synthesized protein in targeted cell populations expressing PGA. The extent of labeling is time-dependent, although appears to be maximal at around 2 hr (*Figure 2B*). This is consistent with a prior study of OPP labeling in cells (*Liu et al., 2012*), and with a scenario that OPP-peptide conjugates eventually undergo turnover. Cellular AF488-azide signal is not uniform across the cell cortex (*Figure 2A*), which is consistent with cell-to-cell variability in PGA expression observed in FLAG immunostained brains (*Figure 2—figure supplement 1C*). When PGA expression was restricted to dopamine neurons using *TH-GAL4*, newly-synthesized proteins are clearly visible in the soma of dopamine neurons within intact brain explants briefly treated with PhAc-OPP, but not in the surrounding tissue (*Figure 2C*). PGA expression targeted to glia via *repo-GAL4*, also confirmed via immunoblot (*Figure 2—figure supplement 1A*), results in visible nascent protein labeling in glial cells (co-labeled with mCherry) upon brief PhAc-OPP treatment, but not in the absence of PhAc-OPP (*Figure 2D*). Hence, POPPi can efficiently label newly-synthesized proteins in neuronal and glial cell populations within the fly brain.

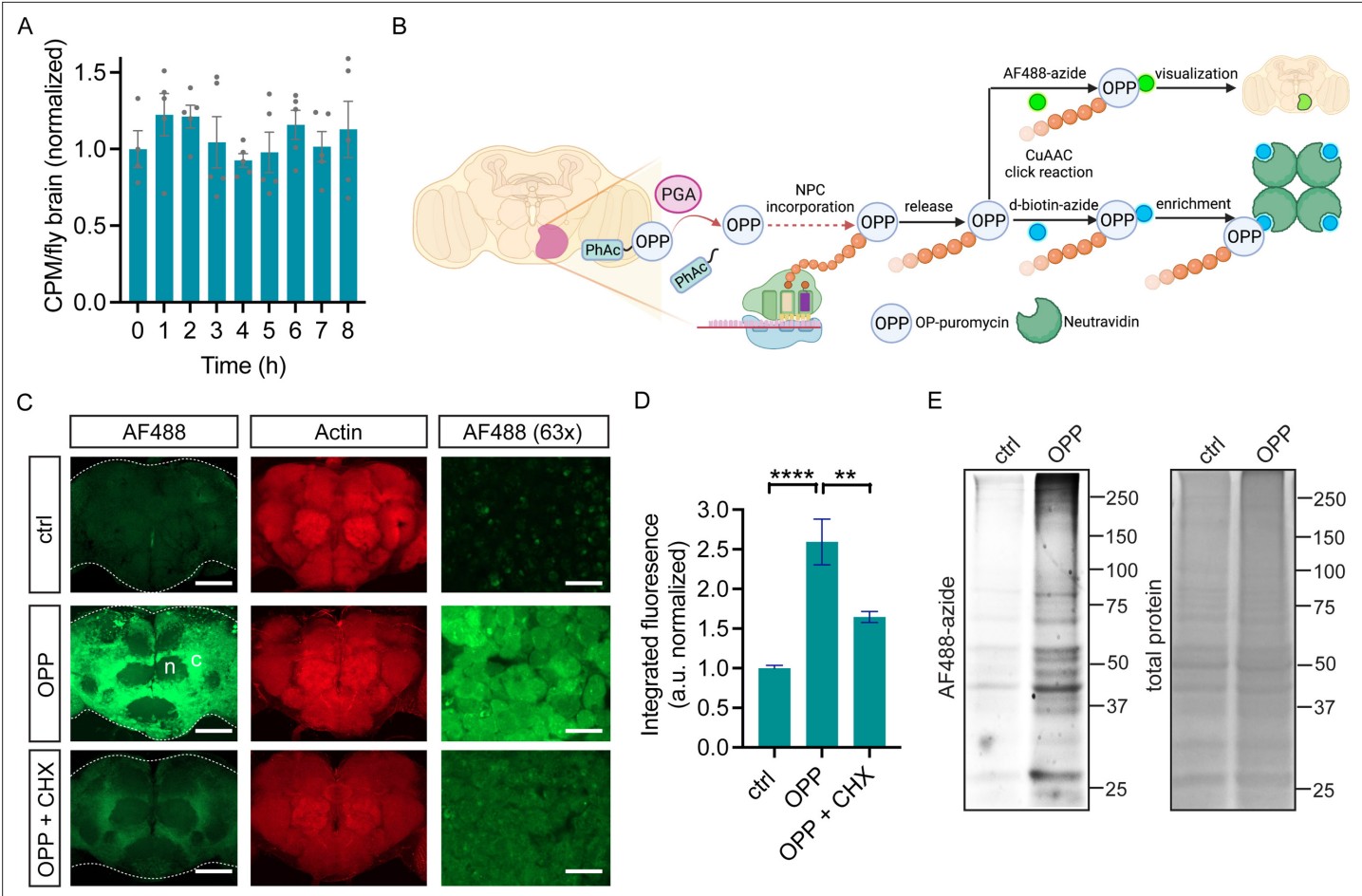

**Figure 1.** O-propargyl-puromycin (OPP) labeling in *Drosophila* brain. (**A**) Global protein synthesis ($^{35}$S-met/cys incorporation) is unaltered for 8 hr in newly-isolated *w$^{1118}$* whole brain preparations (ANOVA, n.s. n=4–5 groups of 8–10 brains per timepoint). (**B**) Schematic illustrating cell type-specific protein synthesis labeling by PGA-dependent OPP incorporation (POPPi) for visualization or capture of the nascent proteome. Spatially targeted penicillin G acylase (PGA) expression catalyzes phenylacetyl-OPP (PhAc-OPP) blocking group removal, liberating OPP for incorporation into nascent polypeptide chains (NPC). OP-puromycylated proteins can be visualized by confocal microscopy following conjugation to a fluorescent-azide or enriched following conjugation to desthiobiotin-azide. (**C**) Newly-synthesized protein (from *w$^{1118}$* brains) visualized by AF488-azide after OPP incubation but not without OPP (ctrl) and diminished signal with cycloheximide (CHX). Labeling appears stronger in cell bodies within the cell cortex (**C**) than in the neuropil (n). Higher magnification (63 x) images are from the cell cortex. Scale bars are 60 µM (or 10 µM for 63 x). (**D**) Quantitation of (**C**) revealing a significant effect of the treatment group (ANOVA, p<0.0001, Bonferroni post-test, **p<0.01, ****p<0.0001, n=8–11 brains/group). (**E**) In-gel fluorescence of brain protein extracts, ctrl is no OPP. Data are mean ± SEM. Schematic in B created on Biorender. See also *Figure 1—figure supplements 1 and 2*.

The online version of this article includes the following source data and figure supplement(s) for figure 1:

**Source data 1.** Source gel images for *Figure 1E*.

**Figure supplement 1.** Penicillin G acylase (PGA)-dependent conversion of phenylacetyl-OPP (PhAc-OPP) to O-propargyl-puromycin (OPP) in *Drosophila*.

**Figure supplement 1—source data 1.** Source western blots for *Figure 1—figure supplement 1C*.

**Figure supplement 2.** O-propargyl-puromycin (OPP) nascent proteome labeling is concentration and time-dependent.

Since OPP incorporation into NPC results in peptide chain termination, we queried whether PhAc-OPP treatment diminishes overall protein synthesis rates. To assess this, we performed $^{35}$S-methionine labeling in parallel to PhAc-OPP treatment of brain explants expressing PGA ubiquitously. We found that PhAc-OPP causes a dose-dependent inhibition of global protein synthesis rates but that this inhibition was minimal up to 100 µM PhAc-OPP (*Figure 2E*) which is the standard concentration used in our labeling experiments. A significant disruption of protein synthesis is only seen above 100 µM PhAc-OPP (*Figure 2E*). Additionally, protein ubiquitination levels are not affected by

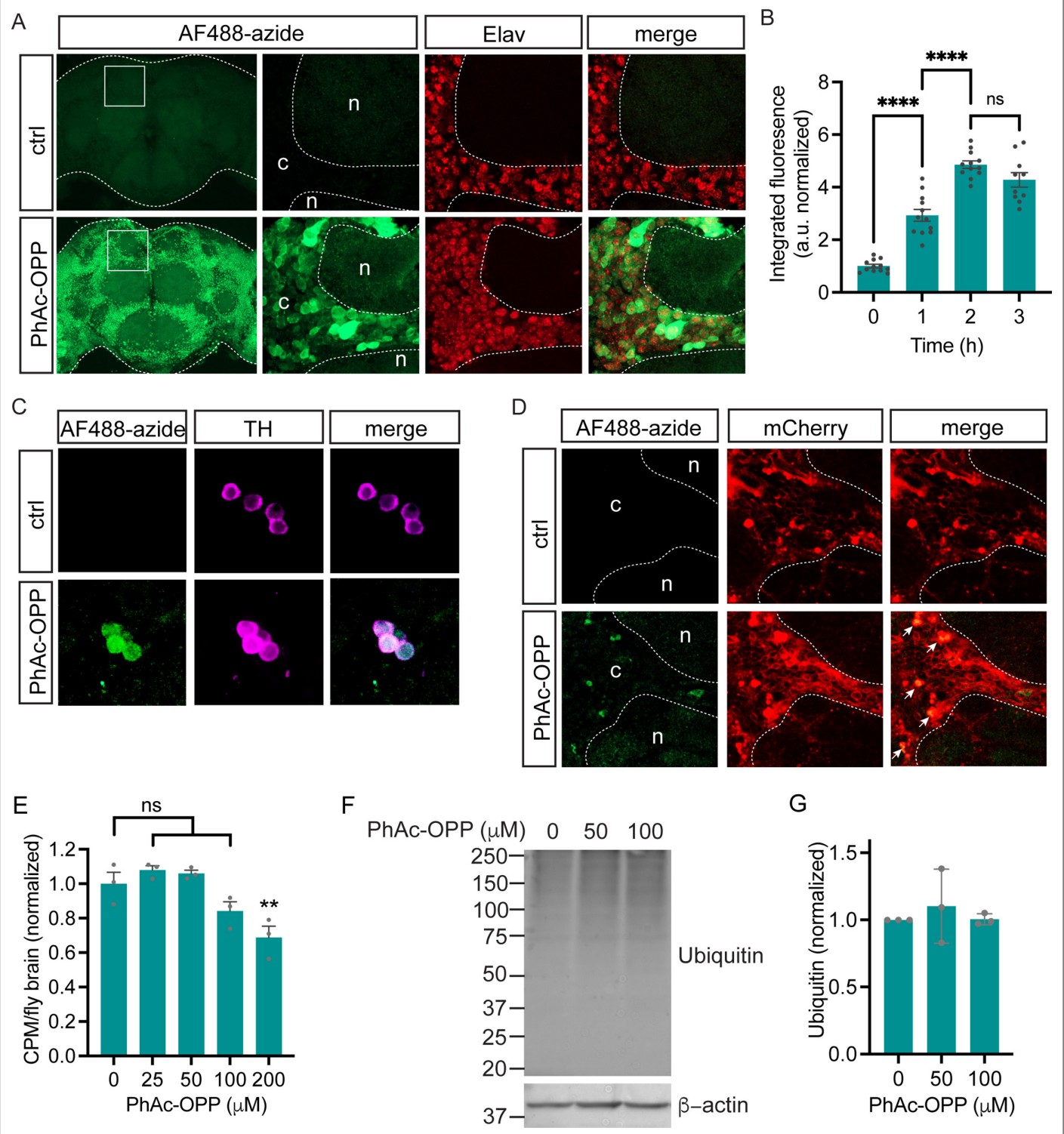

**Figure 2.** Cell type-specific protein synthesis labeling with phenylacetyl-OPP (PhAc-OPP). (**A**) Brains from *elavC155-GAL4;UAS-PGA* flies incubated with PhAc-OPP show widespread protein synthesis labeling in neurons, labeled with Elav neuronal nucleus marker. No labeling in vehicle-treated brains (ctrl). Labeling appears highest in the cell bodies of the cell cortex (c) and minimal in the neuropil (n). (**B**) AF488-azide quantitation after varying durations of PhAc-OPP labeling in pan-neuronal penicillin G acylase (PGA) expressing flies. Significant effect of PhAc-OPP incubation time on labeling (ANOVA, Bonferroni post-test, ****p<0.0001, n=10–12 brains per group). (**C**) Protein synthesis labeling in isolated brains from flies expressing PGA in dopamine neurons (*TH-GAL4/UAS-PGA*). TH, tyrosine hydroxylase. (**D**) Protein synthesis labeling following pan-glial expression of PGA and a membrane-tethered mCherry reporter (*Repo-GAL4, UAS-mCD8::mCherry/UAS-PGA*). Glial cell bodies and neuron-encapsulating surface areas are mCherry-positive. Arrows

*Figure 2 continued on next page*

*Figure 2 continued*

indicate glial cell bodies positive for both mCherry and AF488, indicative of glial protein synthesis labeling. Controls in **A**, **C**, and **D** are vehicle-treated brains. (**E**) Significant effect of 200 µM PhAc-OPP on global protein synthesis in flies expressing PGA ubiquitously via *Actin5C-GAL4* (ANOVA, Bonferroni post-test, **p<0.01, n=3 groups of 8–10 brains/group). (**F**) No significant effect of PhAc-OPP on total protein ubiquitination in flies expressing PGA ubiquitously, quantified in **G** (ANOVA, n=3 groups of 20 brains/group). Data are mean ± SEM. See also *Figure 2—figure supplement 1*.

The online version of this article includes the following source data and figure supplement(s) for figure 2:

**Source data 1.** Source western blots for *Figure 2F*.

**Figure supplement 1.** Cell-specific labeling via penicillin G acylase (PGA) expression.

**Figure supplement 1—source data 1.** Source western blots for *Figure 2—figure supplement 1A*.

incubating brains in up to 100 µM PhAc-OPP for 2 hr (*Figure 2F and G*), consistent with PhAc-OPP not having a major impact on protein turnover under these conditions.

While our primary focus was on labeling brain cell populations, we probed whether PhAc-OPP can penetrate other dissected tissues besides the brain. Rather than express PGA in several individual tissues separately, we drove ubiquitous PGA expression via Actin5C-Gal4 and incubated inverted L3 larvae (which exhibit robust PGA expression (*Figure 1—figure supplement 1*)) in PhAc-OPP, followed by AF488-azide under the same conditions used for labeling and visualizing protein synthesis in isolated brains. We then dissected the fat body, trachea, muscle, and salivary gland for imaging. As seen in the brain, newly-synthesized protein can be clearly visualized in all larval bodily tissues examined (*Figure 3*), suggesting that PhAc-OPP can readily penetrate a variety of tissues besides the brain.

## Rapid cell type-specific nascent proteome capture with PhAc-OPP

As observed upon treating brain explants with OPP (*Figure 1E*), we find that PGA-dependent unblocking of PhAc-OPP in all neurons or in all glia of adult fly brains gives rise to OPP-labeled proteins that span a wide molecular weight range consistent with broad incorporation into the nascent proteome (*Figure 4A and D*). Pan-neuronal PGA expression was coupled to PhAc-OPP treatment and brain lysates were click conjugated to desthiobiotin azide for affinity purification of the neuronal proteome using neutravidin beads. Detection of total biotin-tagged protein with anti-biotin reveals robust enrichment of OPP-labeled protein in pulldown fractions relative to input (whole lysates) (*Figure 4B*). In support of widespread incorporation into NPC, we were able to label specific neuronal proteins of interest. Three crucial synaptic proteins involved in neurotransmitter release, namely Bruchpilot (ERC2 ortholog), Synapsin and Syntaxin, are all substantially enriched in the OPP-labeled neuronal proteome following affinity purification (*Figure 4C*). Similarly, glial PGA expression coupled to PhAc-OPP incubation and affinity purification with desthiobiotin azide yields strong enrichment of desthiobiotin-tagged OPP-labeled protein (*Figure 4E*) and of the glial-specific protein Draper (ortholog of the mammalian engulfment receptor MEGF10) (*Figure 4F*). Conversely, the neuronal SNARE complex protein Syntaxin is not enriched upon pan-glial PGA expression, supporting the conclusion that proteome labeling is restricted to the cell population expressing PGA.

## No effect of cell type-specific PGA expression on adult fitness or survival

Having found that PGA expression coupled to PhAc-OPP incubation can be used to visualize and capture the nascent proteome in a cell type-specific manner, we sought to determine whether PGA expression in flies affects their development, function, or survival. Toward this goal, we crossed *UAS-PGA* flies to pan-neuronal, pan-glial, or ubiquitous *GAL4* driver strains and assessed the resulting progeny. Startle-induced negative geotaxis behavior is dependent on a functionally-intact nervous system and declines progressively with age in flies (*Gargano et al., 2005*; *Martin and Grotewiel, 2006*). Neither pan-neuronal nor pan-glial PGA expression affects negative geotaxis performance across age (*Figure 5A*), while ubiquitous PGA expression causes a pronounced deficit in 3-week-old flies but not at a more advanced age in 6-week-old flies (*Figure 5B*). In accordance with this, ubiquitous PGA expression impairs adult survival, resulting in a ~20% decrease in median lifespan (*Figure 5D* and *Table 1*), whereas no lifespan shortening is seen following pan-neuronal or pan-glial PGA expression (*Figure 5C* and *Table 2*). In fact, lifespan appears to be slightly extended by pan-glial PGA (*Figure 5C* and *Table 2*). Besides its negative effect on function and survival, ubiquitous PGA expression is also

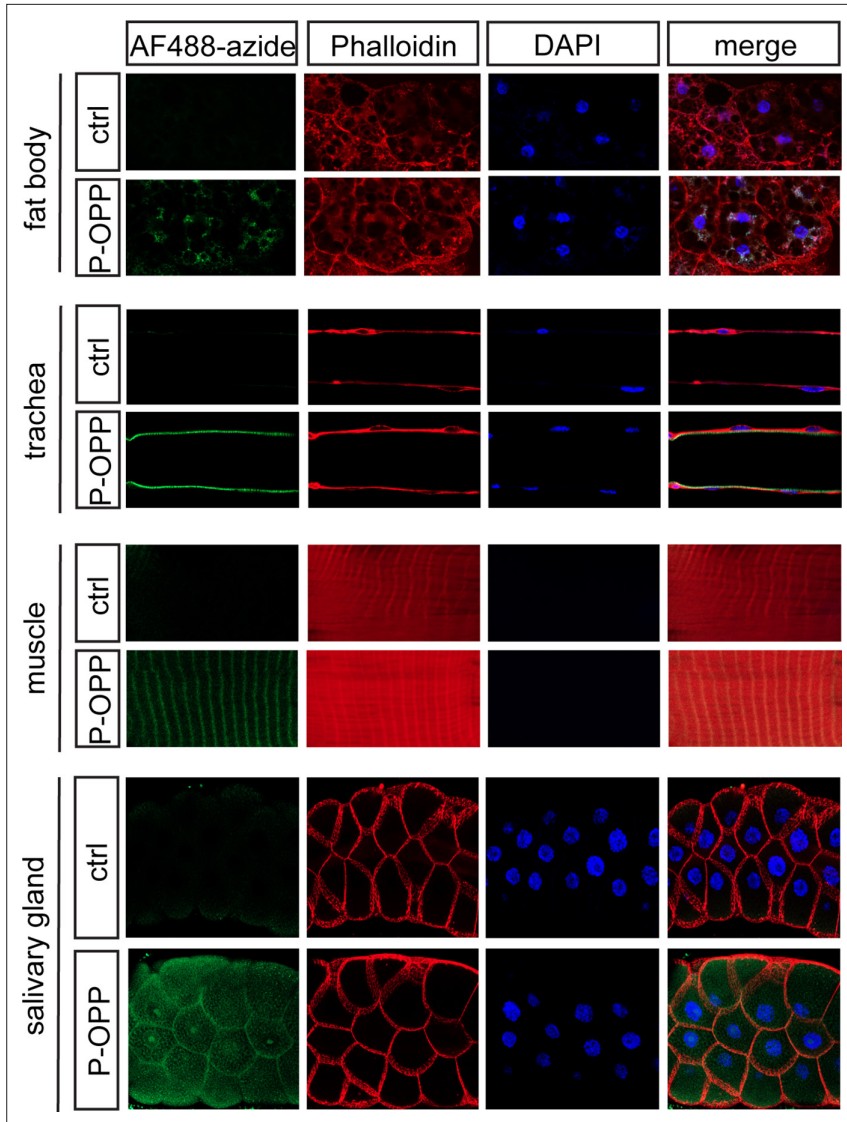

**Figure 3.** Protein synthesis labeling in other major tissues using phenylacetyl-OPP (PhAc-OPP). Tissues from L3 larvae incubated with PhAc-OPP show widespread protein synthesis labeling, visualized following O-propargyl-puromycin (OPP) conjugation to AF488-azide. All larvae expressed penicillin G acylase (PGA) via *Actin5C-Gal4*.

associated with major larval lethality suggesting that development is perturbed, while larval and pupal survival are unaffected by pan-neuronal or pan-glial PGA expression (*Figure 5—figure supplement 1*) and eclosion occurs at the expected Mendelian frequency. Collectively, these data suggest that PGA expression within the brain is well tolerated across the fly lifespan, and raise the possibility that PGA expression in an unknown tissue outside of the nervous system has a negative impact on development, function, and adult survival.

## Brain explant labeling can capture in vivo protein synthesis states

In order to measure cell type-specific protein synthesis under various physiological or pathological conditions, it is important to know whether POPPi can be used to capture in vivo protein synthesis states in newly-isolated brains. To address this question, we examined neuronal protein synthesis labeling in the brains of young and aged flies. It is well established that a widespread age-related decline in protein synthesis occurs in the tissues of numerous organisms including *Drosophila*, when measured across the whole body (*Fleming et al., 1986*) or in heads (*Yang et al., 2019*), and that the decline is due to reduced mRNA translation as well as transcript abundance. To determine whether

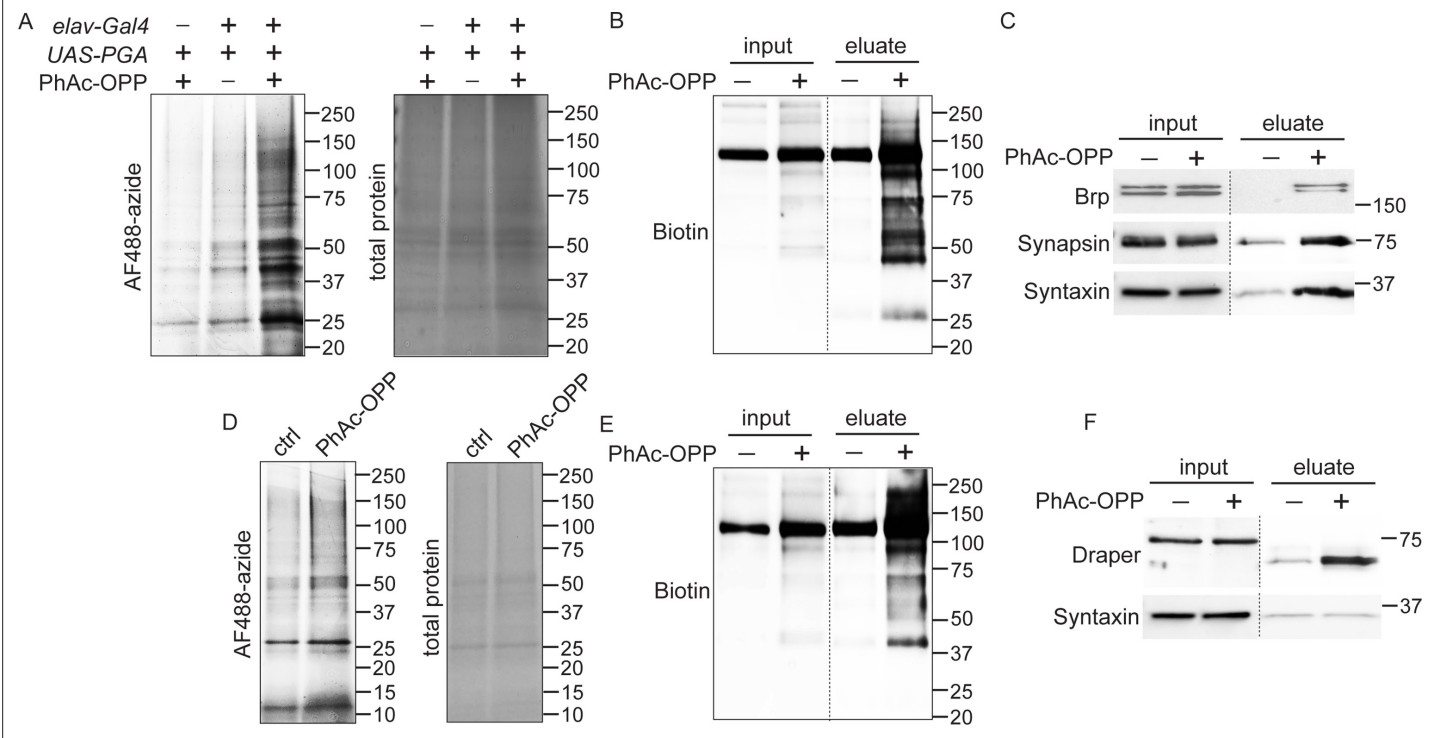

**Figure 4.** Capture and identification of newly-synthesized proteins from neurons and glial. (**A**) In-gel AF488 fluorescence across a broad range of proteins in fly brain extracts expressing penicillin G acylase (PGA) pan-neuronally and following incubation with phenylacetyl-OPP (PhAc-OPP). Controls groups are no PGA expression (with PhAc-OPP) or no PhAc-OPP (with PGA expression). (**B**) Detection of total desthiobiotin-tagged protein in brain lysates (input) or pulldown fractions (eluate) following pan-neuronal PGA expression, PhAc-OPP incubation, conjugation of OPP-labeled protein to desthiobiotin azide and neutravidin bead pulldown. A strong background band of unknown endogenously biotinylated protein is observed between 100–150 kDa. (**C**) Enrichment of neuronal proteins Brp, Synapsin, and Syntaxin following neutravidin pulldown of desthiobiotin-tagged protein from pan-neuronal *elav-GAL4/UAS-PGA* brain extracts. (**D**) Pan-glial PGA expression (*Repo-GAL4/UAS-PGA*) promotes broad O-propargyl-puromycin (OPP) protein labeling in brain extracts following PhAc-OPP incubation but not vehicle control. Enrichment of total desthiobiotin-tagged protein seen following neutravidin bead pulldown of the glial proteome, detected with anti-biotin (**D**) and enrichment of the glial-specific protein Draper (with apparent mol. weight shift between input and eluate) from *Repo-GAL4/UAS-PGA* extracts, but not the neuron-specific protein Syntaxin. Data are representative of three independent experiments.

The online version of this article includes the following source data for figure 4:

**Source data 1.** Source western blots for *Figure 4*.

this same decline is seen specifically in the brains of aging flies, we measured global protein synthesis in young (4-day-old) and aged (21-day-old) wild-type fly brains and observe a substantial decrease in radiolabeled protein from aged fly brain, indicative of decreased protein synthesis (*Figure 6A*). Next, using POPPi in which PGA was expressed pan-neuronally, we were able to observe a decline in neuron-specific bulk protein synthesis in the aging fly brain (*Figure 6B and C*). Strikingly, neuronal protein synthesis was reduced by almost 50% in 3-week-old *elavC155-GAL4/UAS-PGA* flies relative to young (4-day-old) flies of the same genotype (*Figure 6B and C*). Measurement of PGA expression confirmed that this age-related decline in protein labeling was not due to a loss of PGA expression in aged flies (*Figure 6D*). These data support the conclusion that PGA expression coupled with PhAc-OPP incubation can be used to achieve a quantitative assessment of protein synthesis and that an age-dependent decline in protein synthesis can be effectively captured by this method.

## Protein synthesis labeling via PhAc-OPP dietary intake

While incubating brain explants in PhAc-OPP allows rapid protein synthesis labeling in cell populations of interest, we queried whether PhAc-OPP ingested in food can penetrate the brain and label newly-synthesized protein. To assess the feasibility of this approach, we exposed pan-neuronal PGA-expressing flies to various concentrations of PhAc-OPP in sugar-yeast extract food medium for 48 hr

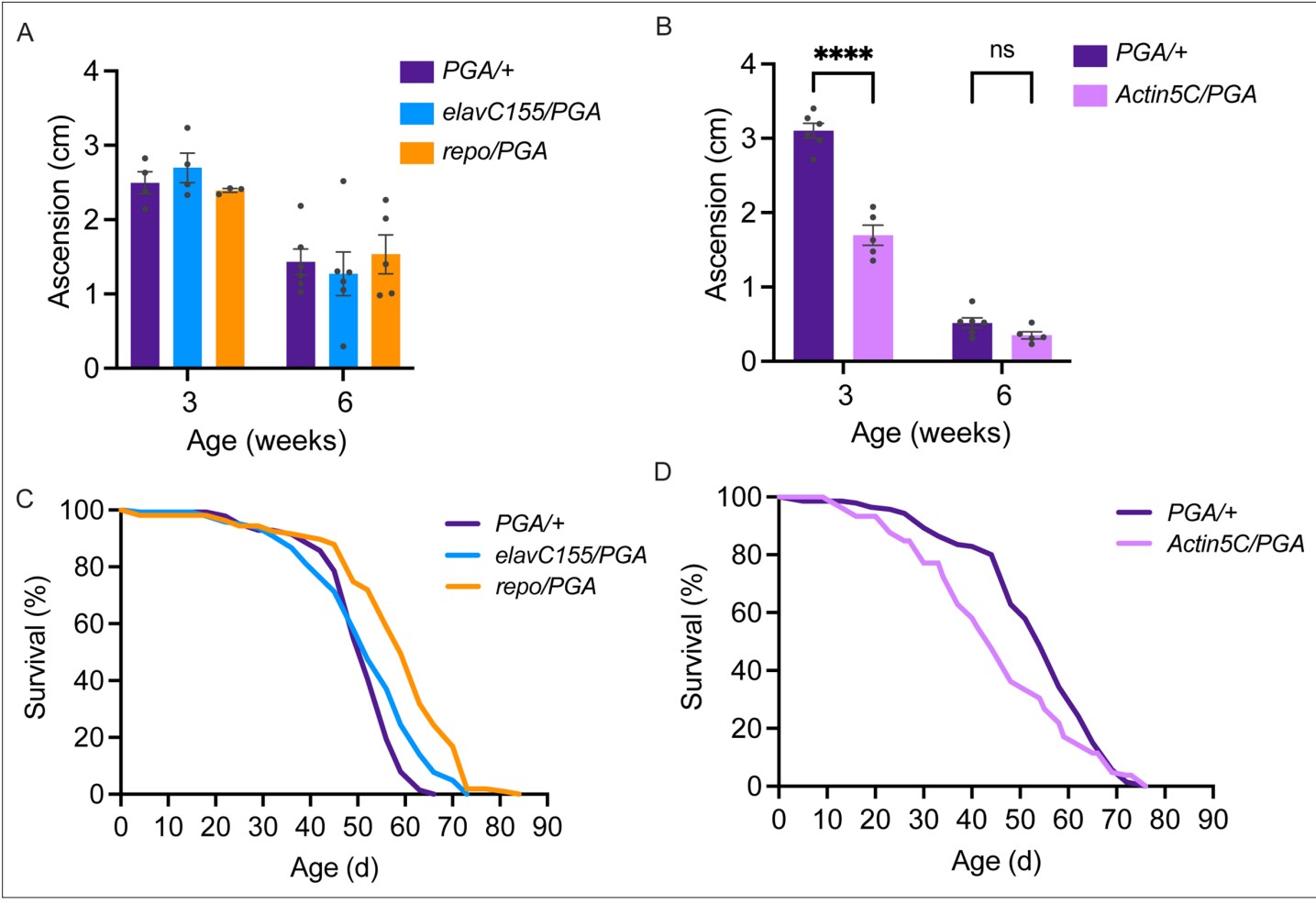

**Figure 5.** No deleterious effect of cell-specific penicillin G acylase (PGA) expression. (**A**) No significant effect of pan-neuronal or pan-glial PGA expression on negative geotaxis behavior at 3 and 6 weeks of age (two-way ANOVA for the effect of age (p<0.0001) and genotype, (n.s.), n=3–6 groups of 25 flies/genotype/age). (**B**) Significant effect of ubiquitous PGA expression on negative geotaxis behavior at 3 weeks of age (two-way ANOVA for the effect of age (p<0.0001) and genotype (p<0.0001), Bonferroni post-test, ****p<0.0001, n=5–6 groups of 25 flies/genotype/age). (**C**) Survival is slightly extended upon pan-neuronal or pan-glial PGA expression. (**D**) Significant effect of ubiquitous PGA expression on survival. For C and D, see *Tables 1 and 2* for experimental n, lifespan metrics, and log-rank (Mantel-Cox) comparison results. Data are mean ± SEM. See also *Figure 5—figure supplement 1*.

The online version of this article includes the following figure supplement(s) for figure 5:

**Figure supplement 1.** Larval lethality following ubiquitous penicillin G acylase (PGA) expression.

then determined whether OPP-labeled nascent protein can be visualized following AF488-azide conjugation. We observe concentration-dependent labeling with PhAc-OPP and that AF488-azide signal becomes significantly higher than background at concentrations ≥1 mM (**Figure 7A and B**). Importantly, flies appear to consume 4 mM PhAc-OPP-containing food in similar quantities to that of control food (**Figure 7C**), suggesting that food consumption is not significantly impaired by the presence of PhAc-OPP, even at the highest dose tested. These results suggest that PhAc-OPP administered

**Table 1.** Ubiquitous penicillin G acylase (PGA) expression effect on survival.

| Genotype | N | Median lifespan (d) | Mean lifespan (d) | Log-rank (vs PGA/+ctrl) |
|---|---|---|---|---|
| *UAS-PGA/+* | 142 | 54 | 53.1 | - |
| *Actin5C-GAL4/UAS-PGA* | 110 | 44 | 45.6 | .006 |

**Table 2.** Neuronal and glial penicillin G acylase (PGA) expression effect on survival.

| Genotype | N | Median lifespan (d) | Mean lifespan (d) | Log-rank (vs. PGA/+ctrl) |
|---|---|---|---|---|
| *UAS-PGA/+* | 152 | 52 | 50.8 | - |
| *elav^C155-GAL4/UAS-PGA* | 147 | 52 | 52.1 | 0.002 |
| *Repo-GAL4/UAS-PGA* | 112 | 63 | 58.4 | <0.001 |

dietarily can penetrate the brain at levels sufficient to obtain detectable albeit subtle nascent protein labeling.

## Discussion

POPPi is a new method to rapidly visualize and capture cell-specific nascent proteomes in whole *Drosophila* brains. This method extends the demonstrated capability of PGA-dependent OPP labeling in cultured neurons (*Barrett et al., 2016*) to proteome labeling within complex nervous system tissue. OPP efficiently labels nascent proteomes within the fly brain and this is substantially blocked by the protein synthesis inhibitor cycloheximide (*Figure 1C and D*), indicating that de novo protein synthesis is key for labeling to occur. We show that our strategy is versatile and can be used to visualize nascent protein synthesis across all CNS neurons, specific to a small population of dopaminergic neurons or limited to glia (*Figure 2*). We also demonstrate that POPPi can be used to capture neuronal or glial proteomes and enrich proteins of interest within those cell populations (*Figure 4*). Global protein synthesis is stable for at least 8 hr in isolated whole brain preparations (*Figure 1A*), supporting an ability to capture in vivo protein synthesis states using this approach. Consistent with this, we were able to detect an age-related decline in bulk neuronal protein synthesis in the fly brain using PhAc-OPP (*Figure 6*). Future efforts will be focused on coupling proteome enrichment to mass spectrometry to interrogate the effects of physiological or pathological stimuli on the proteome at the level of individual proteins.

Spatially-resolved proteomic studies have lagged behind transcriptomic studies in part because issues with limited RNA starting material can be overcome by PCR amplification of cDNA, while no equivalent exists for protein. Nonetheless, flies can be rapidly bred to vast numbers and we anticipate that the facile scalability of *Drosophila* can be leveraged for proteomic studies that focus on

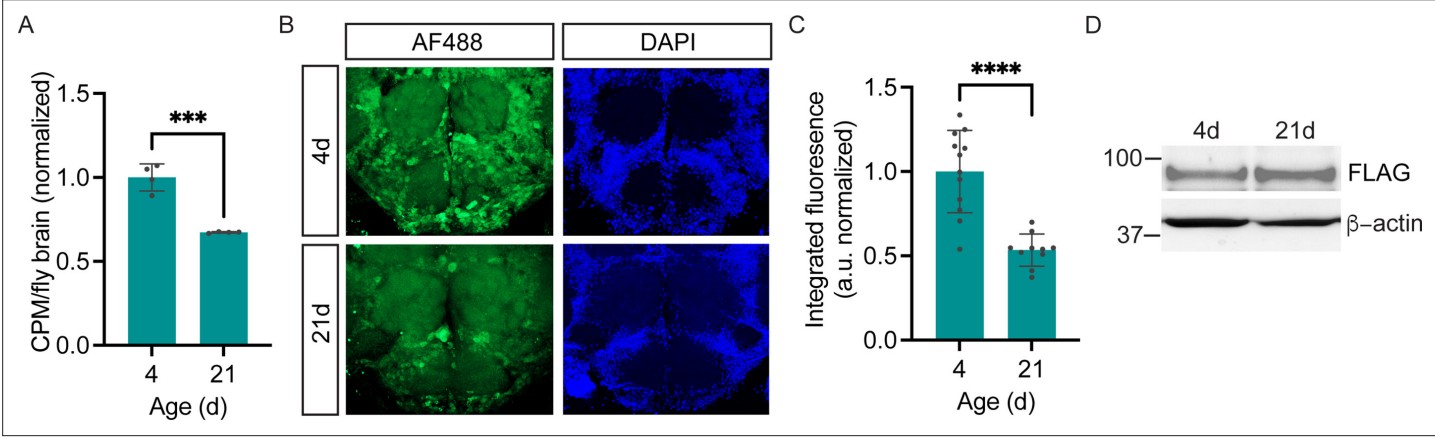

**Figure 6.** Age-dependent decline in neuronal protein synthesis rate. (**A**) Significant age-related decline in whole brain protein synthesis in aging flies, measured by $^{35}$S-met/cys labeling (Student's *t*-test, ***p<0.001, n=4 groups of eight brains/age). (**B**) Nascent proteome labeling in young (4-day-old) vs. aged (21-day-old) fly brains expressing pan-neuronal penicillin G acylase (PGA) following phenylacetyl-OPP (PhAc-OPP) incubation. (**C**) The significant effect of aging on protein synthesis (Student's *t*-test, p<0.0001, n=10–12 brains/group). Fluorescence signals at each age was derived by subtracting from the mean of a no-label control brain population tested in parallel. (**D**) PGA expression is comparable in 4- and 21-day-old *elavC155-GAL4/UAS-PGA* fly brains. Data are mean ± SEM.

The online version of this article includes the following source data for figure 6:

**Source data 1.** Source western blots for *Figure 6D*.

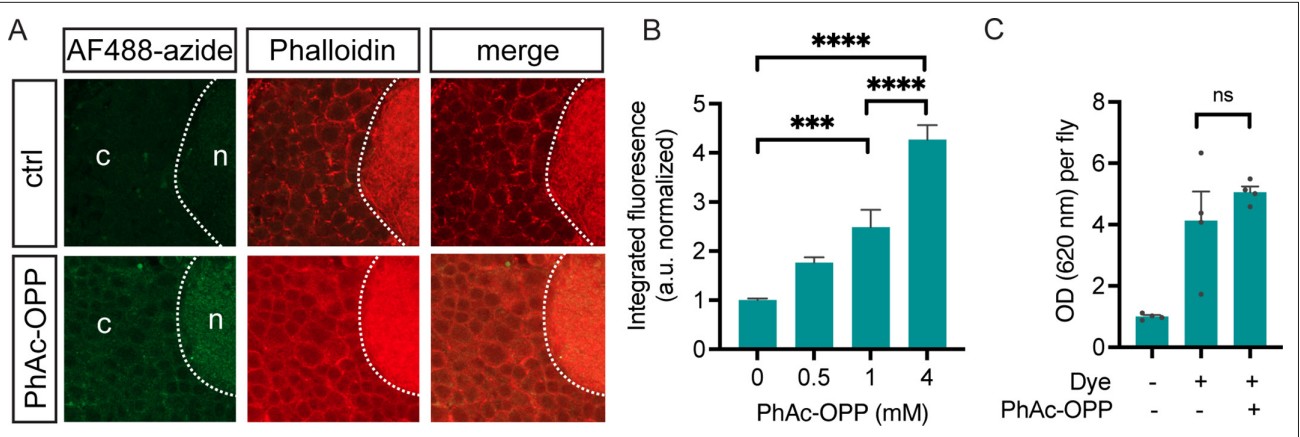

**Figure 7.** Protein synthesis labeling via dietary phenylacetyl-OPP (PhAc-OPP) administration. (**A**) Subtle detection of protein synthesis labeling by AF488-azide following exposure of pan-neuronal penicillin G acylase (PGA)-expressing flies to dietary PhAc-OPP (4 mM) for 48 hr. Brains were counterstained with Dylight Phalloidin-650. (**B**) AF488-azide quantitation after varying concentrations of PhAc-OPP exposure in pan-neuronal PGA-expressing flies. Significant effect of PhAc-OPP concentration on labeling (ANOVA, Bonferroni post-test, ***$p<0.001$, ****$p<0.0001$, n=12–17 brains per group). (**C**) No significant effect of PhAc-OPP (4 mM) on food intake levels over 4 hr (ANOVA, Bonferroni post-test, ns, n=4 groups of eight flies per condition).

small-cell populations. PGA transgenic flies are well-suited for CNS proteomic studies because PGA is robustly expressed in adult, larval, and pupal brains (*Figure 1—figure supplement 1*) as well as via neuronal and glial GAL4 drivers (*Figure 2—figure supplement 1*). Accordingly, while our efforts centered on characterizing proteome labeling within the adult CNS in this study, we anticipate that our chemical genetic approach should be applicable to examining CNS proteomes during fly development. *Drosophila* has been widely used in genetic studies of nervous system development (*Jan and Jan, 2010*), function (*Noyes et al., 2021*), aging (*Piper and Partridge, 2018*), and disease (*Şentürk and Bellen, 2018*). Major insights from these studies highlight strong conservation with mammalian biology and have spurred diverse proteomic analyses of gene expression in the fly nervous system (*Li et al., 2020*; *Owald et al., 2010*; *Mangleburg et al., 2020*; *Wang et al., 2020*) and interest in using flies to model the role of translational regulation in nervous system disease (*Greenblatt and Spradling, 2018*; *Lu et al., 2014*; *Mizielinska et al., 2014*). In addition to the work on diseases such as fragile X syndrome (*Greenblatt and Spradling, 2018*), frontotemporal dementia (*Mizielinska et al., 2014*), and diseases associated with aminoacyl-tRNA synthetase mutations (*Lu et al., 2014*) by others, we recently showed how aberrant translation contributes to neurodegenerative phenotypes caused by the common Parkinson's disease-causing mutation LRRK2 G2019S in *Drosophila* and iPSC-derived dopamine neurons (*Martin et al., 2014a*; *Martin et al., 2014b*; *Kim et al., 2020*), adding to emerging evidence of translational dysregulation in models of Parkinson's disease (*Martin, 2016*). Hence, we foresee many opportunities for PGA-expressing flies to generate insight into nervous system function and disease through cell type-specific proteomic studies. Our study builds on previous findings that OP-puromycin can successfully label cellular protein synthesis when applied to cultured mammalian cells (*Liu et al., 2012*; *Forester et al., 2018*) and from bone marrow cells when administered by i.p. injection to mice (*Hidalgo San Jose and Signer, 2019*). These findings indicate that OP-puromycin can readily penetrate cell membranes and is also able to permeate tissues. PhAc-OPP differs from OP-puromycin by the addition of a phenylacetyl group and *N*- (benzyloxy) carbamate spacer which renders PhAc-OPP more hydrophobic than OP-puromycin. This is anticipated to enhance the ability of PhAc-OPP to penetrate cell membranes by diffusion, while diluted concentrations can still be prepared in an aqueous solution for tissue incubation (see Materials and methods).

One potential concern surrounding the use of puromycin labeling is that its incorporation into NPC causes chain termination, therefore, at high-enough doses, it could impede cellular protein synthesis (*Liu et al., 2012*). This may be circumvented by finding an optimal concentration of puromycin (or analog) which permits sufficient labeling for detection while having minimal impact on total protein synthesis. We assessed this in *Drosophila* brains and found that 100 µM PhAc-OPP allowed us to obtain robust and rapid proteome labeling while not observing significant deficits in global protein

synthesis that were seen at a higher concentration (*Figure 2E*). Another consequence of chain termination when occurring prematurely is the production of truncated protein. Interestingly, we were only able to detect what appears to be full-length protein by western blot following neutravidin enrichment and blotting for individual proteins (*Figure 4C and F*). While truncated proteins may be targeted for degradation and thus depleted from the lysate pool, another simple explanation for this observation is that all antibodies we used which have reported epitopes bind to C-terminal epitopes. These would be missing in truncated proteins, thus precluding their detection using these antibodies.

We assessed whether ectopic PGA expression in fly cell populations perturbs organism development, function, or survival. In contrast to ubiquitous PGA expression, neither neuronal nor glial PGA expression has any discernible negative impact on development, adult function, or survival (*Figure 5* and *Figure 5—figure supplement 1*). Hence, when expressed in individual nervous system cell populations, PGA expression seems to be well tolerated. This may provide an advantage over existing NCAA-based cell-specific labeling strategies, where chronic ANL feeding to flies prior to proteomic assessment significantly impairs fly eclosion as well as negative geotaxis behavior in adults (*Erdmann et al., 2015*). There are additional concerns over how chronic NCAA feeding in this approach might affect protein abundance and overall proteome makeup. For example, the replacement of methionine with the NCAA L-azidohomoalanine (AHA) was seen to cause substantial changes in the abundance of numerous proteins in HeLa cells while AHA incorporation into the developing mouse proteome results in altered expression of about 10% of proteins (*Bagert et al., 2014*; *Calve et al., 2016*). It is also currently unclear whether long-term NCAA administration disrupts eukaryotic metabolism given that AHA and L-homopropargylglycine (HPG) were found to alter global metabolism in *E. coli* (*Steward et al., 2020*). Taken together with the fact that NCAA labeling requires extended dietary methionine depletion, lengthy feeding with amino acid analogs (*Alvarez-Castelao et al., 2017*; *Erdmann et al., 2015*), and that NCAA are not incorporated equally across the proteome, there are significant caveats associated with this labeling strategy that can be avoided using POPPi.

While here we focused on protein synthesis labeling in intact isolated brains, this method should in theory be amenable to proteome labeling in tissues throughout the body, wherever PGA can be adequately expressed. In support of this, we obtained preliminary evidence demonstrating the ability of PhAc-OPP to penetrate and label several bodily tissues (fat body, trachea, muscle, and salivary gland) under the same conditions used for protein synthesis labeling in the brain (*Figure 3*). We also obtained preliminary evidence that PhAc-OPP can penetrate the brain when administered dietarily to intact flies (*Figure 7*). Based on the magnitude of AF488-azide labeling, ingestion at the PhAc-OPP concentration range tested yields lower nascent protein labeling than we were able to achieve in brain explants at much lower concentrations, which we speculate may be due to comparatively lower levels of PhAc-OPP reaching the brain. Future studies will address whether labeling in other tissues can be achieved via dietary PhAc-OPP intake, as this approach may be crucial when tissues cannot be isolated and effectively sustained. Despite this uncertainty, the ability to rapidly quantify protein synthesis in cell populations within the CNS established here opens up many possibilities to address important questions pertaining to nervous system development, function, aging, and disease.

Current proteomic approaches for measuring protein synthesis harbor strengths and weaknesses compared to transcriptomic approaches. Ribosomal profiling and other ribosome capture methods such as TRAP measure ribosomal density on a given transcript which provides a proxy for the rate of protein synthesis but not an actual measure of protein product. Ribosomal profiling has provided important insights into mechanisms of translational control, yet, there are some weaknesses to the method (*Brar and Weissman, 2015*). Perhaps most importantly, inferring protein synthesis rates from a single snapshot of average ribosomal occupancy on a given mRNA is based on assumptions that all ribosomes complete translation, and are not subject to regulated translational pausing or abortion at the time of capture or at any time prior to finishing translation. Additionally, the method itself may miss ribosomal footprints if nuclease digestion is incomplete, and give rise to false readouts of translation from contaminating non-coding RNA fragments. In contrast, assessing translation at the level of synthesized protein, e.g., through quantitative proteomics, should avoid errors associated with inferring protein synthesis rates from ribosomal occupancy, with the caveat that higher sensitivity limits relative to transcriptomic approaches may make transcriptomics the method of choice when starting material is low. We believe a potential major advantage of POPPi over existing methods for measuring cell type-specific protein synthesis is its efficiency, particularly for visualizing protein

synthesis – PhAc-OPP labeling coupled to AF488-azide conjugation can be completed within a few hours (see Methods). Future work will seek to understand the full capabilities and limitations of POPPi, e.g., for profiling rare cell populations in the brain.

In summary, we provide a new labeling method for rapidly visualizing, capturing, and quantifying cell type-specific nascent proteomes within the *Drosophila* brain. We believe this method will be a powerful tool for studying the role of the proteome and translational control in nervous system function with cellular resolution.

# Materials and methods

### Key resources table

| Reagent type (species) or resource | Designation | Source or reference | Identifiers | Additional information |
|---|---|---|---|---|
| Genetic reagent (*D. melanogaster*) | UAS-PGA | This paper | | FLAG-tagged penicillin G acylase under UAS control |
| Genetic reagent (*D. melanogaster*) | Actin5C-Gal4 | Bloomington *Drosophila* Stock Center | BDSC:25374 FLYB:FBti012 7834; RRID:BDSC_ 25374 | FlyBase symbol: P{Act5C-GAL4-w} E1 |
| Genetic reagent (*D. melanogaster*) | Elav(C155)-Gal4 | Bloomington *Drosophila* Stock Center | BDSC:458 FLYB:FBti000 2575; RRID:BDSC_ 458 | FlyBase symbol: P{GawB}elav [C155] |
| Genetic reagent (*D. melanogaster*) | Repo-Gal4 | Bloomington *Drosophila* Stock Center | BDSC:7415 FLYB:FBti001 8692 RRID:BDSC_ 7415 | FlyBase symbol: P{GAL4}repo |
| Genetic reagent (*D. melanogaster*) | TH-Gal4 | Bloomington *Drosophila* Stock Center | BDSC:8848 FLYB:FBti007 2936 RRID:BDSC_ 8848 | FlyBase symbol: P{ple-GAL4.F}3 |
| Biological sample (*D. melanogaster*) | *Drosophila* brain explants | This paper | | Freshly isolated from various *D. melanogaster* genotypes |
| Antibody | Anti-Biotin rabbit polyclonal | Bethyl Laboratories Inc. | Cat: A150109A | 1:1000 |
| Antibody | Anti-Brp (nc82) mouse monoclonal | DSHB | Cat: nc82 | 1:50 |
| Antibody | Anti-Synapsin (3C11) mouse monoclonal | DSHB | Cat: 3C11 | 1:500 |
| Antibody | Anti-Syntaxin (8C3) mouse monoclonal | DSHB | Cat: 8C3 | 1:500 |
| Antibody | Anti-Draper 8A1 mouse monoclonal | DSHB | Cat: 8A1 | 1:400 |
| Antibody | Anti-Draper 5D14 mouse monoclonal | DSHB | Cat: 5D14 | 1:400 |
| Antibody | Anti-Elav 9F8A9 mouse monoclonal | DSHB | Cat: 9F8A9 | 1:100 |
| Antibody | Anti-FLAG (M2) mouse monoclonal | Millipore Sigma | Cat: F1804 | 1:500 |
| Antibody | Anti-FLAG (L5) mouse monoclonal | Novus Biologicals | Cat: NBP1-06712 | 1:1000 |
| Antibody | Anti-Actin-HRP (AC15) mouse monoclonal | Millipore Sigma | Cat: A3854 | 1:2000 |
| Antibody | Anti-GAPDH (GA1R) mouse monoclonal | ThermoFisher | Cat:MA5-15738 | 1:10,000 |
| Antibody | Anti-TH mouse monoclonal | Immunostar | Cat: 22941 | 1:1000 |

*Continued on next page*

*Continued*

| Reagent type (species) or resource | Designation | Source or reference | Identifiers | Additional information |
|---|---|---|---|---|
| Antibody | Anti-Ubiquitin (P4D1) rabbit monoclonal | Cell Signaling Technology | Cat: 3936 | 1:1000 |
| Commercial assay or kit | Click-iT Plus OPP Alexa Fluor 488 Protein Synthesis Assay Kit | ThermoFisher | Cat: C10456 | |
| Chemical compound, drug | PhAc-OPP | This paper and reference (*Barrett et al., 2016*) | | See reference (*Barrett et al., 2016*) for chemical synthesis details |
| Chemical compound, drug | Desthiobiotin azide | Click Chemistry Tools | Cat: 50-210-7822 | |
| Chemical compound, drug | TBTA (Tris[(1-benzyl-1*H*-1,2,3-triazol-4-yl)methyl]amine) | Millipore Sigma | Cat:678937 | |
| Chemical compound, drug | Cu(I)Br | Millipore Sigma | Cat: 61163 | |
| Chemical compound, drug | Neutravidin agarose | Thermo Scientific | Cat: P129204 | |
| Chemical compound, drug | Colloidal blue staining kit | ThermoFisher | Cat: LC6025 | |

## *Drosophila* stocks and culture

PGA-expressing flies were generated by subcloning N-terminal FLAG-tagged full-length PGA cDNA (gift of C. Doe) into the fly transformation vector pUAST between KpnI and EcoRI restriction sites. After sequence verification and successful construct expression following transient transfection of *UAS-PGA* and *GAL4* into *Drosophila* S2 cells, the construct was microinjected into $w^{1118}$ fly embryos (Best-gene, Inc). Transgenic FLAG-PGA expression was confirmed by FLAG Western blotting of adult head extracts after crossing all generated *UAS-PGA* lines to *Actin5C-GAL4* (ubiquitous), *elavC155-GAL4* (pan-neuronal), and *repo-GAL4* (pan-glial). The following strongest-expressing lines were used throughout the study: *UAS-PGA-1* for ubiquitous expression and *UAS-PGA-3* for neuronal or glial expression. The *repo-GAL4, UAS-mCD8::mCherry* recombinant line was a gift from M. Freeman and all other lines were obtained from the Bloomington *Drosophila* Stock Center: *TH-GAL4* (line 8848); *elav^{C155}-GAL4* (line 458); *repo-GAL4* (line 7415); *Act5c-GAL4* (line 25374). All flies were reared and aged at 25 °C/60% relative humidity under a 12 hr light-dark cycle on a standard food medium.

## PhAc-OPP synthesis and preparation

PhAc-OPP was synthesized from chemical precursors, analyzed by proton NMR, analytical thin-layer chromatography, and purified by flash chromatography as previously described (*Barrett et al., 2016*).

To prepare PhAc-OPP for tissue incubations, a stock solution (20 mM dissolved in DMSO) was diluted to 100 µM in Schneider's *Drosophila* medium (with L-glutamine and sodium bicarbonate), bath sonicated for 6 min, vortexed continuously for 30 s prior to adding the proteasomal inhibitor MG132 (60 µM) and then kept at RT until use. To prepare PhAc-OPP for ingestion in a small food volume, 2 X the final desired concentration of PhAc-OPP was dissolved in 0.25 ml of distilled water via several brief tip sonication pulses, briefly heated to 55 °C then mixed thoroughly in a 1:1 volumetric ratio with 0.25 ml of 2 X sugar-yeast food solution (3% agar/10% sucrose/20% yeast extract) previously heated to 100 °C and cooled to 55 °C. After the food had solidifed, the microfuge tube containing the food was cut just above the food surface and inserted upright into a cotton plug placed at the bottom of a shell vial. Flies were transferred to the vial and kept in a humidified chamber throughout feeding for the indicated durations.

## Adult brain preparations and $^{35}$S-methionine/cysteine assessment of de novo protein synthesis

Adult brains were isolated and maintained for protein synthesis assessment in Schneider's *Drosophila* medium (formulated for *Drosophila* cells and tissues and supplemented with L-glutamine and sodium bicarbonate), based on previously reported methods (*Ayaz et al., 2008*; *Gibbs and Truman, 1998*). Prior long-term culture of *Drosophila* CNS explants has often included the addition of insulin and/or serum to the culture medium (*Ayaz et al., 2008*; *Gibbs and Truman, 1998*). As global protein

synthesis was maintained at a constant rate for 8 hr of ex vivo culture in the absence of exogenously added insulin and serum (*Figure 1A*) and as insulin/serum might artificially stimulate protein synthesis, we omitted them for our short-term rapid protein labeling approach. Adult fly brains (8–10 per genotype) were harvested in Schneider's medium and transferred to Schneider's medium containing $^{35}$S-methionine/cysteine (2 mCi/ml) to metabolically label newly-synthesized protein. Brains were incubated for 30 min at 25 °C with gentle orbital shaking, washed twice in 1 ml of PBS then flash frozen. Brains were homogenized in modified RIPA extraction buffer (50 mM Tris-HCl pH 7.4, 150 mM NaCl, 100 mM EGTA, 1% NP-40, 0.1% SDS, protease inhibitor cocktail) on ice using a pestle gun. After a 30 min incubation on ice, lysates were centrifuged at 14,000 × g for 15 min and the supernatant was retained. Protein was precipitated by the addition of methanol and heparin (lysate:heparin (100 mg/ml):methanol volume ratio of 150:1.5:600), centrifuged at 14,000 × g for 2 min, the supernatant was removed, and the pellet was air dried. The protein pellet was resuspended in 8 M urea/150 mM Tris, pH 8.5, and $^{35}$S-methionine/cysteine incorporation was measured by liquid scintillation counting (counts per minute, CPM) and normalized to the number of brains.

### Effect of PhAc-OPP on global protein synthesis

Adult fly brains (8–10 per genotype) from the standard laboratory control strain $w^{1118}$ were harvested in Schneider's medium and transferred to Schneider's medium containing $^{35}$S-methionine/cysteine (2 mCi/ml) and PhAc-OPP at the indicated concentrations. Brains were incubated for 2 hr at 25 °C with gentle orbital shaking, washed twice in 1 ml of PBS then flash frozen. Thawed brains were homogenized and processed for protein precipitation and measurement of $^{35}$S incorporation by scintillation counting as described above.

### Rapid OPP/PhAc-OPP labeling of newly-synthesized CNS proteins

To enable rapid labeling of nascent CNS proteins, adult *Drosophila* brain explants maintained in Schneider's *Drosophila* medium were incubated with OPP or PhAc-OPP within 30 min of collection ('newly-isolated brains'). Brains (~15 for immunocytochemistry, 100 for enrichment, and immunoblotting) were harvested in Schneider's medium and then immediately transferred to Schneider's medium containing OPP (50 µM unless otherwise indicated) or PhAc-OPP (100 µM unless otherwise indicated) and MG-132 (60 µM) for 2 hr (unless otherwise indicated) at 25 °C with gentle orbital shaking. Brains were washed twice briefly in 1 ml PBS and then immediately processed for immunocytochemistry or enrichment and immunoblot detection as described below.

### Immunocytochemical detection of OP-puromycylated protein

OP-puromycylated tissues were fixed for 20 min in 4% paraformaldehyde in PBS-T (PBS, pH 7.4 containing 0.3% Triton-X-100) at RT with gentle rocking, washed once in chilled PBS-T (5 min) and once in chilled PBS (5 min). Detection of OP-puromycylated protein was achieved by conjugation to a fluorophore-azide called AF488 picolyl azide (AF488-azide) via CuAAC click reaction. A click working reagent was prepared containing TBTA (200 µM), freshly prepared Cu(I)Br (0.5 mg/ml), AF488-azide (0.1 X concentration), and 1 X concentration OPP reaction buffer (the last two reagents derived from the Click-iT Plus OPP Alexa Fluor 488 Protein Synthesis Assay Kit (Invitrogen) and at concentrations relative to those recommended in the manufacturer's instructions). The click working reagent was vortexed for 30 s then applied to brains and incubated for 30 min at 25 °C with gentle orbital shaking and protection from light. Tissues were washed briefly with 1 ml of Click-iT Reaction Rinse Buffer and either whole-mounted or additionally counterstained with Dylight Phalloidin-650 or immunostained for cell markers or DAPI stained prior to mounting. Tissues were imaged on a Zeiss LSM900 confocal microscope. For in-gel fluorescence assays, OPP/PhAc-OPP treated brains were lightly fixed (4% PFA/10 min), click-conjugated to AF488-azide as described above then lysed in de-crosslinking buffer (300 mM Tris-HCl/2% SDS/protease inhibitors) for 2 hr at 60 °C prior to SDS-PAGE, gel fixation (40% methanol/10% acetic acid), 30 min washing in ddH$_2$O, AF488-azide detection and subsequently total protein visualization using colloidal blue stain.

### Immunoblot detection of enriched OP-puromycylated protein

The protocol is modified from *Marter et al., 2019*. Brains were thawed on ice and homogenized in 100 µL of homogenization buffer (0.5% SDS/PBS containing 2 X Complete EDTA-free Protease Inhibitor

Cocktail). Samples were incubated on ice for 20 min with mixing every 5 min, at 95 °C for 5 min, then on ice for 5 min. Triton-X-100 was added to a final concentration of 0.2% and pre-equilibrated neutra-vidin agarose (25 µL/sample) was added to pre-clear lysates of endogenous biotinylated proteins via sample incubation on a nutator at 4 °C for 1 hr. Samples were centrifuged (3000 × g, 5 min, 4 °C) and to the supernatant, click chemistry reagents were added as follows: TBTA (final concentration 200 µM), desthiobiotin azide (20 µM), freshly prepared Cu(I)Br (0.5 mg/ml). On each addition, samples were vortexed for 10 s. Samples were incubated overnight on a nutator at 4 °C, centrifuged at 3000 × g for 5 min/4 °C then the supernatant was diluted in PBS with 2 X protease inhibitor cocktail to a final volume of 300 µl and desalted (Zeba spin desalting columns). Total protein concentration in lysates was determined by BCA assay to standardize input for neutravidin agarose enrichments. Neutravidin agarose was equilibrated in 1% NP-40/PBS for three washes and 1% NP-40 was added to desalted lysates for 20 min on ice. After removing a portion of the lysate for input analysis, samples were added to equilibrated neutravidin agarose (50 µL/sample) and incubated overnight at 4 °C on a nutator. Samples were centrifuged (3000 × g, 5 min, 4 °C) and washed in 1% NP-40/2 X protease inhibitor cocktail/PBS five times than in PBS for three additional washes, with 2 min of end-over-end mixing at each wash step. Desthiobiotin-conjugated protein was eluted from neutravidin agarose by incubating in 8 mM biotin for 1 hr in a thermomixer at 25 °C/1200 rpm shaking. Laemmli buffer was then added to eluate and input for downstream analysis by immunoblotting.

## Rapid PhAc-OPP labeling of larval tissues for immunocytochemical detection

Wandering L3 larvae were washed briefly in PBS and then inverted to expose bodily tissues to the surrounding solution for labeling. Inverted larval preps were immediately transferred to Schneider's medium, then incubated with 100 µM PhAc-OPP and 60 µM MG-132 followed by PFA fixation and click conjugation to AF488-azide under the same conditions used for adult fly brains. Tissues targeted for imaging (fat body, trachea, muscle, and salivary glands) were separated, counterstained with Dylight Phalloidin-650, and mounted for imaging on a Zeiss LSM 900 confocal microscope.

## Adult survival

Adult females (0–3 days of age, 100–150 flies per genotype, selecting against newly-eclosed flies) were collected under brief anesthesia and transferred to fresh food vials at 25 flies per vial. Flies were then transferred to fresh food vials every 3–4 days throughout the experiment and dead or censored (escaped or stuck in food) flies were counted during each transfer to fresh food.

## Negative geotaxis behavior

Cohorts of 75–100 female flies (0–3 days old, selecting against flies with visible signs of recent eclosion) were collected under brief anesthesia and transferred to fresh food vials to recover (25 flies/vial). Flies were aged for 6 weeks with transfer to fresh food twice per week. On the day of testing, flies were transferred to empty vials, allowed 1 min to rest and then tapped to the bottom of the vial three times within a 1 s interval to initiate climbing. The position of each fly was captured in a digital *Figure 4* second after climbing initiation. Automated image analysis was performed using the particle analysis tool on Scion Image to derive x–y coordinates for each fly thus providing the height climbed, as previously described (*Gargano et al., 2005*). The performance of flies in a single vial was calculated from the average height climbed by all flies in that vial to generate a single datum (N=1). Performance of each line was then derived from the average scores of 5–6 vials tested for the line (N=5–6).

## Food intake

Blue dye (erioglaucine disodium salt) was added at 1.5% w/v to the food during preparation to measure food consumption as previously described (*Chittoor-Vinod et al., 2020*). Briefly, flies were housed for 4 hr on dye-containing food with or without the presence of 4 mM PhAc-OPP, flash frozen in liquid nitrogen, washed in chilled water, lysed in chilled water, centrifuged (15 min/13,000 rpm/4 °C) and supernatants were assessed for OD 620 nm (SpectraMax i3x). OD readings were normalized to the number of flies in the lysate.

## Western blotting

Samples were electrophoresed on 4–20% Tris-Glycine gradient gels and transferred to nitrocellulose membrane for immunoblotting using the following antibodies:

From the Developmental Studies Hybridoma Bank: Brp (nc82) 1:50; Synapsin-1 (3C11) 1:500; Syntaxin (8C3) 1:500; Drpr (1:1 mix of 5D14:8A1) 1:400; Repo (8D12) 1:200. Other antibodies used were biotin (Bethyl Laboratories) 1:1000; FLAG (M2) (Sigma) 1:500; TH (Immunostar) 1:1000; Actin-HRP (Sigma) 1:1000; Ubiquitin (P4D1) (Cell Signaling Technology) 1:1000; GAPDH (GA1R) (ThermoFisher Scientific) 1:10,000.

## Statistical analysis

Quantified data are mean ± SEM and individual data points are plotted for data with n<10. Sample sizes for time-course experiments were determined based on evidence from pilot experiments. Statistical analysis details for each individual experiment are described in figure legends, including the number of flies or the number of groups of flies used ($n$), statistical tests (unpaired two-tailed Student's $t$-test, ANOVA or two-way ANOVA), and Bonferroni post hoc analysis with associated $p$ values. All statistical analyses were performed using GraphPad Prism except Log-rank (Mantel-Cox) tests for lifespan comparisons performed in SPSS.

## Data availability

All data generated or analyzed during this study are included in the manuscript's main Figures or Figure supplements. Source data are provided for Western blots and gels.

## Materials availability

Newly created *Drosophila* line (*UAS-PGA*) has been deposited at the Bloomington *Drosophila* Stock Center (stock ID pending) and will be shared upon request.

## Acknowledgements

We thank Richard Goodman and Chris Doe for initial discussions on the project, Chris Doe and Sen-Lin Lai for providing PGA cDNA, the OHSU Advanced Light Microscopy Core for microscope use, and the OHSU Medicinal chemistry core for PhAc-OPP synthesis. The following antibodies were obtained from the Developmental Studies Hybridoma Bank, created by the NICHD of the NIH and maintained at The University of Iowa: Brp nc82 and Synapsin 3C11 (developed by E Buchner), Syntaxin 8C3 (S Benzer and N Colley), Draper 8A1 and 5D14 (M Logan) and Elav 9F8A9 (G Rubin). This work was funded by OHSU Neurology Foundation Funds (IM).

## Additional information

### Funding

| Funder | Grant reference number | Author |
|---|---|---|
| OHSU Foundation | | Ian Martin |

The funders had no role in study design, data collection and interpretation, or the decision to submit the work for publication.

### Author contributions

Stefanny Villalobos-Cantor, Conceptualization, Formal analysis, Validation, Investigation, Visualization, Methodology, Writing - original draft, Writing – review and editing; Ruth M Barrett, Conceptualization, Formal analysis, Investigation, Methodology, Writing – review and editing; Alec F Condon, Conceptualization, Formal analysis, Validation, Investigation, Methodology, Writing – review and editing; Alicia Arreola-Bustos, Formal analysis, Investigation, Writing – review and editing; Kelsie M Rodriguez, Investigation; Michael S Cohen, Conceptualization, Resources, Methodology, Writing – review and editing; Ian Martin, Conceptualization, Formal analysis, Supervision, Funding acquisition, Validation, Investigation, Visualization, Methodology, Writing - original draft, Project administration, Writing – review and editing

### Author ORCIDs

Alec F Condon (iD) http://orcid.org/0000-0003-2655-2121

Kelsie M Rodriguez (iD) http://orcid.org/0000-0002-0821-6717
Michael S Cohen (iD) http://orcid.org/0000-0002-7636-4156
Ian Martin (iD) http://orcid.org/0000-0002-5912-1777

**Decision letter and Author response**
Decision letter https://doi.org/10.7554/eLife.83545.sa1
Author response https://doi.org/10.7554/eLife.83545.sa2

## Additional files

### Supplementary files
• MDAR checklist

### Data availability
All data generated or analyzed during this study are included in the manuscript and supporting files.

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
