## [Editor Report]

The authors developed a versatile labeling strategy to allow visualization and identification of newly synthesized proteins in a cell population of interest. The approach enables cell-specific nascent proteome labeling from brain tissues to examine the role of translational control in different physiological and pathological states.

---

## [Decision Letter]

**Decision letter after peer review:**

Thank you for submitting your article "Rapid Cell Type-Specific Nascent Proteome Labeling in *Drosophila*" for consideration by *eLife*. Your article has been reviewed by 3 peer reviewers, and the evaluation has been overseen by a Reviewing Editor and K VijayRaghavan as the Senior Editor. The reviewers have opted to remain anonymous.

The authors describe a potentially useful method to quantitatively identify and visualize the nascent proteome of individual cell populations. The characterization of the approach is incomplete and preliminary, and the method needs to be better validated.

Essential revisions:

1. Test if PhAc-OPP penetrates into various non-brain dissected tissues, such as muscle, fat body, and trachea. Etc. This could be done in larvae by dissecting and inverting larvae, and exposing various tissues to the PhAc-OPP in solution.

2. Test if ingested PhAc-OPP penetrates into the adult brain expressing elav>PGA.

What are the pros/cons of this method versus currently available methods based on ribosome profiling? The sensitivity of this method would be good to comment on in comparison with existing methods. Also, when would this be the method of choice over other methods?

3. How available are the reagents needed for POPPi, and are they easy to obtain by any research lab?

4. The reagent AF488-azide is referred to differently in different places in the manuscript and was not well introduced or described.

5. Because OPP incorporation is sequence-independent, it should be able to label and terminate a peptide chain at any amino acid residue. Therefore I would expect the western blot to show a wider band or a smear after OPP or POPPi labeling. However, all the blots shown in the paper show similarly sharp bands as in the corresponding controls. Can the authors provide an explanation? Are the shorter (incomplete) peptides being degraded very quickly?

6. Many of the issues raised by reviewer 3 are important and should be addressed.

In essence, the characterization of the method needs to be extended and validated.

*Reviewer #1 (Recommendations for the authors):*

To address my main criticism in the public review, I suggest the following experiments.

1. Test if PhAc-OPP penetrates into various non-brain dissected tissues, such as muscle, fat body, and trachea. Etc. This could be done in larvae by dissecting and inverting larvae, and exposing various tissues to the PhAc-OPP in solution.

2. Test if ingested PhAc-OPP penetrates into the adult brain expressing elav>PGA.

*Reviewer #2 (Recommendations for the authors):*

I would like the authors to address the following questions:

What are the pros/cons of this method versus currently available methods based on ribosome profiling? The sensitivity of this method would be good to comment on in comparison with existing methods. Also, when would this be the method of choice over other methods?

How available are the reagents needed for POPPi, and are they easy to obtain by any research lab?

The reagent AF488-azide is referred to differently in different places in the manuscript and was not well introduced or described.

Because OPP incorporation is sequence-independent, it should be able to label and terminate a peptide chain at any amino acid residue. Therefore I would expect the western blot to show a wider band or a smear after OPP or POPPi labeling. However, all the blots shown in the paper show similarly sharp bands as in the corresponding controls. Can the authors provide an explanation? Are the shorter (incomplete) peptides being degraded very quickly?

*Reviewer #3 (Recommendations for the authors):*

1. It is not clear what percentage of the newly synthesized polypeptides incorporate puromycin which would allow visualization or immunopurification of the protein through the use of azide conjugates. Since the inclusion of puromycin in the nascent peptide prevents further elongation of the nascent peptide, reaching the levels of puromycin to saturate the cells would likely be detrimental. On the other hand, when lower amounts of puromycin are used, it is likely that the lower expressed genes/polypeptides will not be labelled. In any case, the authors should estimate the ratio of the newly synthesized targeted by puromycin. This can be done by using PhAc-OPP in elav-Gal4 UAS-PGA brains in combination with azide-biotin and S35 methionine. This would allow immunopurification with an antibody specific for a protein (such as bruchpilot), measuring the level of S35, and a second immunopurification with streptavidin and measuring again with S35. This can allow the estimation of the percentage of newly synthesized proteins that are targeted by PhAc-OPP.

2. It is not clear how there can be clear full-size bands in the Western blots in figure 3. The addition of PhAc-OPP should truncate the protein at random sites that correspond to where the OPP is added to the nascent peptide. This should result in smears in principle unless there is a tendency of OPP to get incorporated towards the end of the protein. An explanation should be included in the discussion to help the reader understand the data better. This also undermines the argument that the fact that there are bands of different sizes suggests that the proteome is uniformly targeted. Since puromycin addition truncates the gene products at random positions, the band sizes should not be informative and indicative of unbiased labelling of the proteome.

3. Although the imaging using OPP and Alexa488-azide looks rather uniform labeling in figure 1C when PhAc-OPP is used in combination with elav-Gal4 or repo-Gal4 UAS-PGA, the Alexa488-azide signal is much more salt and pepper distributed in figure 2A and D (especially in the magnified insets). This is especially prominent in glia where very few of the mCherry expressing cells are labelled by Alexa488-azide in Figure 2D. This suggests that the proteome of some cells will be much higher represented in the samples when azide-biotin is used to immunopurified proteins expressed by a cell type, severely limiting the use of the technique.

4. The fact that OPP-mediated labeling is quite uniform suggests that the translation levels do not change very much between cells and the reason for the non-uniform signal in figure2 may result from the non-uniform expression of UAS-PGA. The authors should stain the elav-Gal4 or repo-Gal4 UAS-PGA with an anti-Flag antibody to check for non-uniform expression of PGA.

5. In figure 1 figure supplement 2B the authors should test incubation times that are between 0-2 hours better since at 2 hours the system already seems to be saturated.

In summary, the technique needs to be better characterized to be more usable by the fly community. In addition, the demonstration experiments to show that there is an age-dependent decrease in translation rates are not the most informative demonstration of the use of the technique. Hence, this manuscript needs major revision and a better demonstration of the use of the technique to warrant publication in *eLife*.

[Editors’ note: further revisions were suggested prior to acceptance, as described below.]

Thank you for resubmitting your work entitled "Rapid Cell Type-Specific Nascent Proteome Labeling in *Drosophila*" for further consideration by *eLife*. Your revised article has been evaluated by K VijayRaghavan (Senior Editor) and a Reviewing Editor.

The manuscript has been improved but there are some remaining issues that need to be addressed, as outlined by the reviewer. It is really important to address these concerns in detail. Please can you do this at the earliest and submit a revised manuscript?

*Reviewer #1 (Recommendations for the authors):*

The authors satisfied the major concerns in my original review. However, I have an additional major concern.

The data presented do not convince me that POPPi is a robust and useful method for capturing and identifying newly-synthesized proteins in a cell type-specific manner (e.g. w/ biotin-azide). In the response to my concern of my original review, the authors suggest that background bands seen in negative control lanes of western blots (e.g. now Figure 4A) are due to non-specific AF488-azide incorporation into all proteins (newly synthesized + old). This is unexpected to me, as I had assumed the click chemistry was specific to OPP-labeled proteins. Regardless, assuming that non-OPP-labeled proteins are labeled by AF488-azide, I interpret the western blots in Figure 4A and Figure 4C as showing weak 'signal above background'. This is especially concerning because Figure 4A,C were performed by labeling a large portion of the cells within the collected tissue (pan-neuron and pan-glia). So, labeling in smaller subsets of cell types (e.g. dopaminergic neurons) would be expected to result in even lower 'signal above background'. I outline below additional thoughts and suggestions that could help address my concerns.

Recommendations for the authors:

1. To more accurately assess the signal-to-background of cell type-specific labeled proteins, the authors should provide a proper negative control in Figure 4A and 4C. The "vehicle-only" negative control may underestimate the background levels of AF488-azide incorporation. I would argue that the most appropriate negative control is to incubate wild-type brains (e.g. elav>GFP) with PhAc-OPP. Hypothetically, PhAc-OPP could be uncaged by endogenous fly enzymes, which would result in higher background levels than the vehicle-only control. As far as I can tell, the possibility of uncaging PhAc-OPP without PGA is not discussed and remains untested in this manuscript. Furthermore, line 203 "PGA-dependent unblocking of PhAc-OPP…" is thus not supported by their data using the "vehicle-only" negative control in Figure 4. I suggest that the authors repeat the data in Figure 4A and 4C using the appropriate negative control I mentioned.

2. The authors should provide western blots similar to Figures 4A,C, but using the biotin-azide labeling reagent. Since biotin-azide enables researchers to capture and detect OPP-labeled proteins using streptavidin-bead pulldown, it is critical that researchers understand the global signal-to-background when using this reagent. I suggest that the authors repeat Figure 4A,C using biotin-azide to label and streptavidin-fluorophore to detect. Furthermore, since *Drosophila* contains naturally biotinylated proteins, these new western blots would provide the fairest representation of signal-to-background. For example, naturally biotinylated proteins will bind to the streptavidin beads and thus be background on western blots and in Mass spectrometry (MS) data.

3. I would like to see MS data to support the data presented in Figure 4 B,D. These figures show that POPPi can enrich cell type-specific labeled proteins. However, the background bands for Synapsin, Syntaxin, and Draper are concerning and suggest the sensitivity of this method is low. To convince me that there is global signal over background, I would like to see MS analysis of samples in Figure 4B,D. For example, MS of streptavidin-pulldown from the following brain genotypes (1) elav>PGA + PhAc-OPP, (2) repo>PGA + PhAc-OPP, (3) elav>GFP + PhAc-OPP. Using quantitative approaches such as TMT labeling or SAINT analysis, I would expect to see enrichment of Draper to glia and Syntaxin to neurons, for example.

*Reviewer #2 (Recommendations for the authors):*

I'm generally satisfied with the responses and changes to the manuscript. My only remaining request is that the authors include a line for PhAc-OPP in the Key Resources Table. Clearly, this is a key reagent, and the authors should be transparent that it is something that needs to be synthesized, which will likely limit how widespread the method is used by the community.

---

## [Author Response]

Essential revisions:1. Test if PhAc-OPP penetrates into various non-brain dissected tissues, such as muscle, fat body, and trachea. Etc. This could be done in larvae by dissecting and inverting larvae, and exposing various tissues to the PhAc-OPP in solution.

We thank the reviewer for suggesting this approach to assessing PhAc-OPP penetration into other tissues besides brain. We inverted L3 larvae expressing PGA via the *Actin5C-Gal4* driver, and then immediately incubated inverted larvae in 100 µM PhAc-OPP for 2h under the same conditions as previously used for labeling brain tissue. Our prior results indicate robust PGA expression in larvae from this genotype (Figure 1 —figure supplement 1). Upon dissecting and imaging individual tissues (fat body, trachea, muscle and salivary gland), we observed readily detectable protein synthesis labeling. This indicates that PhAc-OPP is able to penetrate a variety of tissues besides the brain to label newly-synthesized protein. These results are presented in the revised manuscript (Figure 3), Results (line 190), Discussion (line 367) and Methods (line 542).

2. Test if ingested PhAc-OPP penetrates into the adult brain expressing elav>PGA.What are the pros/cons of this method versus currently available methods based on ribosome profiling? The sensitivity of this method would be good to comment on in comparison with existing methods. Also, when would this be the method of choice over other methods?

To assess the feasibility of protein synthesis labeling via dietary PhAc-OPP intake, we exposed pan-neuronal PGA-expressing flies (*elav-Gal4>UAS-PGA*) to various concentrations of PhAc-OPP in sugar/yeast extract food for 48h then determined whether OPP-labeled nascent protein is detectable following AF488-azide conjugation. We observe concentration-dependent labeling with PhAc-OPP and AF488azide signal becomes significantly higher than background at concentrations ≥1 mM (Figure 7A and 7B). Importantly, flies appear to consume 4 mM PhAc-OPP-containing food in similar quantities to that of control food (Figure 7C), suggesting that food consumption is not significantly impaired by the presence of PhAc-OPP, even at the highest dose tested. These results suggest that PhAc-OPP administered dietarily can penetrate the brain at levels sufficient to obtain detectable albeit subtle nascent protein labeling. These results are described in the Results (line 265), Discussion (line 370) and Methods (line 438).

We addressed questions related to the pros/cons, sensitivity etc. of our method in new text added to the manuscript Discussion as follows:

“Current proteomic approaches for measuring protein synthesis harbor strengths and weaknesses compared to transcriptomic approaches. Ribosomal profiling and other ribosome capture methods such as TRAP measure ribosomal density on a given transcript which provides a proxy for the rate of protein synthesis but not an actual measure of protein product. Ribosomal profiling has provided important insights into mechanisms of translational control, yet, there are some weaknesses to the method (45). Perhaps most importantly, inferring protein synthesis rates from a single snapshot of average ribosomal occupancy on a given mRNA is based on assumptions that all ribosomes complete translation, and are not subject to regulated translational pausing or abortion at the time of capture or at any time prior to finishing translation. Additionally, the method itself may miss ribosomal footprints if nuclease digestion is incomplete, and give rise to false readouts of translation from contaminating non-coding RNA fragments. In contrast, assessing translation at the level of synthesized protein, e.g. through quantitative proteomics, should avoid errors associated with inferring protein synthesis rates from ribosomal occupancy, with the caveat that sensitivity limits relative to transcriptomic approaches may make transcriptomics the method of choice when starting material is low. We believe a potential major advantage of POPPi over existing methods for measuring cell type-specific protein synthesis is its efficiency, particularly for visualizing protein synthesis – PhAc-OPP labeling coupled to AF488-azide conjugation can be completed within a few hours (see Methods). Future work will seek to understand the full capabilities and limitations of POPPi, e.g. for profiling rare cell populations in the brain.”

3. How available are the reagents needed for POPPi, and are they easy to obtain by any research lab?

PhAc-OPP is not currently commercially available, yet its chemical synthesis has been previously described in detail (Reference 12) which labs can use to have the compound synthesized, e.g. through a university chemistry core or a company. All other compounds used for the methods described are commercially available, including AF488-azide for visualizing protein synthesis and desthiobiotin-azide for enriching labeled protein. All reagent details are provided in the key resources table. PGA transgenic flies will be made available upon request, as indicated in the manuscript.

4. The reagent AF488-azide is referred to differently in different places in the manuscript and was not well introduced or described.

We now consistently refer to this reagent in its short form (AF488-azide) throughout the manuscript except when introducing the reagent in the Methods, where we added the statement “Detection of OP-puromycylated protein was achieved by conjugation to a fluorophore-azide called AF488 picolyl azide (AF488-azide) via CuAAC click reaction”. We regret the prior lack of clarity.

5. Because OPP incorporation is sequence-independent, it should be able to label and terminate a peptide chain at any amino acid residue. Therefore I would expect the western blot to show a wider band or a smear after OPP or POPPi labeling. However, all the blots shown in the paper show similarly sharp bands as in the corresponding controls. Can the authors provide an explanation? Are the shorter (incomplete) peptides being degraded very quickly?

It is likely that truncated proteins would be targeted for eventual degradation and depleted from the protein pool as the reviewer indicates. However, another simple explanation is that commercially available antibodies used for Brp, Synapsin and Draper are all reported to recognize epitopes at the C-terminal end of these proteins, which would be absent in most truncated protein and therefore not detectable on Western blot. The epitope for Syntaxin has not been mapped. Hence, the depletion of truncated proteins combined with an inability to detect them using the available antibodies used in this study likely explain their absence of western blots. We added this interpretation to the Discussion (line 337).

6. Many of the issues raised by reviewer 3 are important and should be addressed.

Please see our point-by-point response to reviewer 3 comments below.

In essence, the characterization of the method needs to be extended and validated.

Our new findings presented in the revised manuscript support the capability of our method to (i) label newly-synthesized protein in *Drosophila* tissues besides the brain and (ii) label newly-synthesized protein in the brain when the labeling compound PhAc-OPP is administered to flies dietarily. We believe these revisions extend characterization of the method substantially.

Reviewer #1 (Recommendations for the authors):To address my main criticism in the public review, I suggest the following experiments.1. Test if PhAc-OPP penetrates into various non-brain dissected tissues, such as muscle, fat body, and trachea. Etc. This could be done in larvae by dissecting and inverting larvae, and exposing various tissues to the PhAc-OPP in solution.

We thank the reviewer for suggesting this approach to assessing PhAc-OPP penetration into other tissues besides brain. We inverted L3 larvae expressing PGA via the *Actin5C-Gal4* driver, and then immediately incubated inverted larvae in 100 µM PhAc-OPP for 2h under the same conditions as previously used for labeling brain tissue. Our prior results indicate robust PGA expression in larvae from this genotype (Figure 1 —figure supplement 1). Upon dissecting and imaging individual tissues (fat body, trachea, muscle and salivary gland), we observed readily detectable protein synthesis labeling. This indicates that PhAc-OPP is able to penetrate a variety of tissues besides the brain to label newly-synthesized protein. These results are presented in the revised manuscript (Figure 3), Results (line 190), Discussion (line 367) and Methods (line 542).

2. Test if ingested PhAc-OPP penetrates into the adult brain expressing elav>PGA.

To assess the feasibility of protein synthesis labeling via dietary PhAc-OPP intake, we exposed pan-neuronal PGA-expressing flies (*elav-Gal4>UAS-PGA*) to various concentrations of PhAc-OPP in sugar/yeast extract food for 48h then determined whether OPP-labeled nascent protein is detectable following AF488-azide conjugation. We observe concentration-dependent labeling with PhAc-OPP and AF488azide signal becomes significantly higher than background at concentrations ≥1 mM (Figure 7A and 7B). Importantly, flies appear to consume 4 mM PhAc-OPP-containing food in similar quantities to that of control food (Figure 7C), suggesting that food consumption is not significantly impaired by the presence of PhAc-OPP, even at the highest dose tested. These results suggest that PhAc-OPP administered dietarily can penetrate the brain at levels sufficient to obtain detectable albeit subtle nascent protein labeling. These results are described in the Results (line 265), Discussion (line 370) and Methods (line 438).

Reviewer #2 (Recommendations for the authors):I would like the authors to address the following questions:What are the pros/cons of this method versus currently available methods based on ribosome profiling? The sensitivity of this method would be good to comment on in comparison with existing methods. Also, when would this be the method of choice over other methods?

To address these questions, the following text has been added to the Discussion (line 386):

“Current proteomic approaches for measuring protein synthesis harbor strengths and weaknesses compared to transcriptomic approaches. Ribosomal profiling and other ribosome capture methods such as TRAP measure ribosomal density on a given transcript which provides a proxy for the rate of protein synthesis but not an actual measure of protein product. Ribosomal profiling has provided important insights into mechanisms of translational control, yet, there are some weaknesses to the method (45). Perhaps most importantly, inferring protein synthesis rates from a single snapshot of average ribosomal occupancy on a given mRNA is based on assumptions that all ribosomes complete translation, and are not subject to regulated translational pausing or abortion at the time of capture or at any time prior to finishing translation. Additionally, the method itself may miss ribosomal footprints if nuclease digestion is incomplete, and give rise to false readouts of translation from contaminating non-coding RNA fragments. In contrast, assessing translation at the level of synthesized protein, e.g. through quantitative proteomics, should avoid errors associated with inferring protein synthesis rates from ribosomal occupancy, with the caveat that sensitivity limits relative to transcriptomic approaches may make transcriptomics the method of choice when starting material is low. We believe a potential major advantage of POPPi over existing methods for measuring cell type-specific protein synthesis is its efficiency, particularly for visualizing protein synthesis – PhAc-OPP labeling coupled to AF488azide conjugation can be completed within a few hours (see Methods). Future work will seek to understand the full capabilities and limitations of POPPi, e.g. for profiling rare cell populations in the brain.”

How available are the reagents needed for POPPi, and are they easy to obtain by any research lab?

PhAc-OPP is not currently commercially available, yet its chemical synthesis has been previously described in detail (Reference 12) which labs can use to have the compound synthesized, e.g. through a university chemistry core or a company. All other compounds used for the methods described are commercially available, including AF488-azide for visualizing protein synthesis and desthiobiotin-azide for enriching labeled protein. All reagent details are provided in the key resources table. PGA transgenic flies will be made available upon request, as indicated in the manuscript.

The reagent AF488-azide is referred to differently in different places in the manuscript and was not well introduced or described.

We now consistently refer to this reagent in its short form (AF488-azide) throughout the manuscript except when introducing the reagent in the Methods, where we added the statement “Detection of OP-puromycylated protein was achieved by conjugation to a fluorophore-azide called AF488 picolyl azide (AF488-azide) via CuAAC click reaction”. We regret the prior lack of clarity.

Because OPP incorporation is sequence-independent, it should be able to label and terminate a peptide chain at any amino acid residue. Therefore I would expect the western blot to show a wider band or a smear after OPP or POPPi labeling. However, all the blots shown in the paper show similarly sharp bands as in the corresponding controls. Can the authors provide an explanation? Are the shorter (incomplete) peptides being degraded very quickly?

It is likely that truncated proteins would be targeted for eventual degradation and depleted from the protein pool as the reviewer indicates. However, another simple explanation is that commercially available antibodies used for Brp, Synapsin and Draper are all reported to recognize epitopes at the C-terminal end of these proteins, which would be absent in most truncated protein and therefore not detectable on Western blot. The epitope for Syntaxin has not been mapped. Hence, the combined depletion of truncated proteins in addition to an inability to detect them using the available antibodies used in this study likely explain their absence of western blots. We added this interpretation to the Discussion (line 337).

Reviewer #3 (Recommendations for the authors):1. It is not clear what percentage of the newly synthesized polypeptides incorporate puromycin which would allow visualization or immunopurification of the protein through the use of azide conjugates. Since the inclusion of puromycin in the nascent peptide prevents further elongation of the nascent peptide, reaching the levels of puromycin to saturate the cells would likely be detrimental. On the other hand, when lower amounts of puromycin are used, it is likely that the lower expressed genes/polypeptides will not be labelled. In any case, the authors should estimate the ratio of the newly synthesized targeted by puromycin. This can be done by using PhAc-OPP in elav-Gal4 UAS-PGA brains in combination with azide-biotin and S35 methionine. This would allow immunopurification with an antibody specific for a protein (such as bruchpilot), measuring the level of S35, and a second immunopurification with streptavidin and measuring again with S35. This can allow the estimation of the percentage of newly synthesized proteins that are targeted by PhAc-OPP.

This is conceptually an interesting approach, although we believe there are major practical constraints that render this approach and the suggested experiment unlikely to provide quantitatively meaningful results. First, IP would likely not work well since the available antibodies to our assessed cell type-specific proteins bind to Cterminal epitopes, thus missing any truncated protein fragments and therefore likely a substantial portion of labeled newly-synthesized protein. Second, streptavidin enrichment is expected to enrich all labeled cellular protein, not just the protein-ofinterest, and hence, the ^35^S signal for any single protein-of-interest would have to be distinguished from all the other proteins in the eluate. This is potentially feasible by combining autoradiography with western blot, yet due to the antibody issue mentioned above, we would again likely miss any truncated labeled protein present.

2. It is not clear how there can be clear full-size bands in the Western blots in figure 3. The addition of PhAc-OPP should truncate the protein at random sites that correspond to where the OPP is added to the nascent peptide. This should result in smears in principle unless there is a tendency of OPP to get incorporated towards the end of the protein. An explanation should be included in the discussion to help the reader understand the data better. This also undermines the argument that the fact that there are bands of different sizes suggests that the proteome is uniformly targeted. Since puromycin addition truncates the gene products at random positions, the band sizes should not be informative and indicative of unbiased labelling of the proteome.

Please see response to the last comment of Reviewer 2.

3. Although the imaging using OPP and Alexa488-azide looks rather uniform labeling in figure 1C when PhAc-OPP is used in combination with elav-Gal4 or repo-Gal4 UAS-PGA, the Alexa488-azide signal is much more salt and pepper distributed in figure 2A and D (especially in the magnified insets). This is especially prominent in glia where very few of the mCherry expressing cells are labelled by Alexa488-azide in Figure 2D. This suggests that the proteome of some cells will be much higher represented in the samples when azide-biotin is used to immunopurified proteins expressed by a cell type, severely limiting the use of the technique.

To clarify, glial cell bodies populate the brain cell cortex at much lower numbers than neurons, but cortex glia form an extensive network around their surrounding neurons with each glial cell encapsulating many neurons. This gives the membrane-tethered mCherry signal a honeycomb-like appearance where the majority of mCherry signal is not from glial cell bodies but from glial membrane extensions that spread across the neuronal network (Figure 2D). While these membrane extensions do not appear AF488-azide labeled, most of the solidly-filled glial cell bodies do, consistent with predominant cell body protein synthesis labeling we observe with neurons and with unblocked OPP. However, we acknowledge that there appears to be more overall cellto-cell variability in generally cellular AF488-azide signal when brains are incubated in PhAc-OPP compared to unblocked OPP, although there is also some variability in labeling with unblocked OPP (Figure 1C and Figure 1 —figure supplement 2), which is not dependent on PGA-induced unblocking. This suggests that variability in protein synthesis labeling may be partly due to cell-to-cell differences in global translation, and partly due to differences in PGA expression (see below).

4. The fact that OPP-mediated labeling is quite uniform suggests that the translation levels do not change very much between cells and the reason for the non-uniform signal in figure2 may result from the non-uniform expression of UAS-PGA. The authors should stain the elav-Gal4 or repo-Gal4 UAS-PGA with an anti-Flag antibody to check for non-uniform expression of PGA.

We performed FLAG immunostaining of *elav-Gal4>UAS-PGA* adult brains and observed non-uniform FLAG levels between cells suggesting a degree of cell-to-cell variability in PGA expression. The data are presented in Figure 2 —figure supplement 1 and described in the Results (line 168).

5. In figure 1 figure supplement 2B the authors should test incubation times that are between 0-2 hours better since at 2 hours the system already seems to be saturated.

Indeed, at 2h of labeling with unblocked OPP labeling, AF488-azide signal is close to maximal and based on this OPP result, we focused on shorter time points (e.g. 1 hour) with PhAc-OPP (Figure 2B). We did not pursue additional time points with unblocked OPP, since PhAc-OPP is the main focus of our study.

[Editors’ note: further revisions were suggested prior to acceptance, as described below.]

Reviewer #1 (Recommendations for the authors):The authors satisfied the major concerns in my original review. However, I have an additional major concern.The data presented do not convince me that POPPi is a robust and useful method for capturing and identifying newly-synthesized proteins in a cell type-specific manner (e.g. w/ biotin-azide). In the response to my concern of my original review, the authors suggest that background bands seen in negative control lanes of western blots (e.g. now Figure 4A) are due to non-specific AF488-azide incorporation into all proteins (newly synthesized + old). This is unexpected to me, as I had assumed the click chemistry was specific to OPP-labeled proteins. Regardless, assuming that non-OPP-labeled proteins are labeled by AF488-azide, I interpret the western blots in Figure 4A and Figure 4C as showing weak 'signal above background'. This is especially concerning because Figure 4A,C were performed by labeling a large portion of the cells within the collected tissue (pan-neuron and pan-glia). So, labeling in smaller subsets of cell types (e.g. dopaminergic neurons) would be expected to result in even lower 'signal above background'. I outline below additional thoughts and suggestions that could help address my concerns.

We are pleased that our resubmitted manuscript satisfied major concerns in the Reviewer’s original review.

Regarding the additional point raised, it is important to note that the presence of background AF488-azide binding to protein is mainly restricted to brain lysates as we do not see significant background in control groups assessed in confocal imaged wholemount brains (Figure 2A, C and D) or other fly tissues (Figure 3) incubated in PhAcOPP then in AF488-azide. We would also like to point out that these gels show total brain lysates, i.e. there is no enrichment for labeled protein. As indicated in our original response, our primary intended application for the AF488-azide reagent is to visualize protein synthesis by confocal imaging whole-mount brains (as in Figures 2A, C and D). The data in previous Figures 4A and 4C were only intended to show that protein synthesis visualized in brains imaged by confocal microscopy was not merely derived from just a few labeled proteins, but from a range of proteins that span the molecular weight range of the gel.

In order to address the Reviewer’s concern, we have performed additional experiments to incorporate the requested control group for in-gel fluorescence assays and also to show total OPP-labeled protein from neuronal and glial proteomes by the biotin tagging and pulldown method, as described more below. Our new data clearly demonstrates strong signal-to-background for cell type-specific protein synthesis labeling by PhAcOPP, that matches the strong signal-to-background already seen in imaging data from fly brain and other tissues. This, coupled to data showing our ability to enrich for cell type-specific proteins following labeling in targeted cell populations provides rigorous quality control for our method and demonstrates its clear utility for achieving robust cell type-specific protein synthesis labeling.

Recommendations for the authors:1. To more accurately assess the signal-to-background of cell type-specific labeled proteins, the authors should provide a proper negative control in Figure 4A and 4C. The "vehicle-only" negative control may underestimate the background levels of AF488-azide incorporation. I would argue that the most appropriate negative control is to incubate wild-type brains (e.g. elav>GFP) with PhAc-OPP. Hypothetically, PhAc-OPP could be uncaged by endogenous fly enzymes, which would result in higher background levels than the vehicle-only control. As far as I can tell, the possibility of uncaging PhAc-OPP without PGA is not discussed and remains untested in this manuscript. Furthermore, line 203 "PGA-dependent unblocking of PhAc-OPP…" is thus not supported by their data using the "vehicle-only" negative control in Figure 4. I suggest that the authors repeat the data in Figure 4A and 4C using the appropriate negative control I mentioned.

The negative control indicated by the Reviewer was included in the manuscript, not for the data in Figure 4, but for confocal imaging of protein synthesis in whole-mount brains. Specifically, we showed the absence of protein synthesis labeling when brains are incubated with PhAc-OPP but the PGA transgene is not expressed (see Figure 2—figure supplement 1b; data are for flies harboring UAS-PGA transgene but no GAL4 driver, therefore no PGA expression). Hence, we had previously addressed the possibility of PGA-independent PhAc-OPP uncaging and shown via the control group in Figure 2—figure supplement 1b that there is no substantial uncaging of PhAc-OPP without PGA expression.

However, to fully address any potential concerns of the Reviewer, we repeated the in-gel fluorescence assay incorporating the additional control group requested with brains incubated in PhAc-OPP but not expressing PGA (Figure 4A). For this control group, we chose flies that harbor the UAS-PGA transgene but no GAL4 driver, hence they are unable to express PGA. Exactly as seen with confocal brain imaging, this control group does not show any substantive protein synthesis labeling visualized by A488-azide signal (Figure 4A), again consistent with a lack of PhAc-OPP uncaging in the absence of PGA expression. We have added this new data as new Figure 4A and also added text to the Results (line 164) in order to describe this control group more clearly in the manuscript.

2. The authors should provide western blots similar to Figures 4A,C, but using the biotin-azide labeling reagent. Since biotin-azide enables researchers to capture and detect OPP-labeled proteins using streptavidin-bead pulldown, it is critical that researchers understand the global signal-to-background when using this reagent. I suggest that the authors repeat Figure 4A,C using biotin-azide to label and streptavidin-fluorophore to detect. Furthermore, since *Drosophila* contains naturally biotinylated proteins, these new western blots would provide the fairest representation of signal-to-background. For example, naturally biotinylated proteins will bind to the streptavidin beads and thus be background on western blots and in Mass spectrometry (MS) data.

We performed the experiments requested in order to observe total biotin-tagged protein signal from neurons or glia following pulldown with neutravidin beads. The approach described by the reviewer (biotin-azide capture of OPP-labeled protein followed by pulldown with avidin beads) is the same approach we had used in blots to detect enrichment of individual neuron-specific or glial-specific proteins (now Figures 4C and F), hence we used these samples to blot for global biotin-tagged protein as requested by the Reviewer. We used anti-biotin to do this instead of streptavidinfluorophore suggested by the Reviewer, as anti-biotin is well validated for western blots on fly tissue (Reference (12)). Using this approach, we observe strong biotin-tagged protein signal in eluted fractions from PhAc-OPP-treated brains and very little background in the vehicle-treated control group likely corresponding to endogenously biotinylated proteins present in flies (new Figures 4B and E). Biotin-tagged protein can also be seen in inputs (whole lysates) for PhAc-OPP-treated brains and this is weaker than in eluted fractions, as would be expected. As the reviewer indicates, naturally biotinylated proteins are present in flies and can manifest as background on western blots. Consistent with this, we find one major non-specific band across all groups and present in both input and eluates. Hence, these new data demonstrate very high signalto-background for enriched biotin-tagged proteins. We believe these data address the concern about signal strength in in-gel fluorescence assays as pulldown of biotin tagged protein is a preferred method for assessing global labeling signal, as indicated by the Reviewer.

3. I would like to see MS data to support the data presented in Figure 4 B,D. These figures show that POPPi can enrich cell type-specific labeled proteins. However, the background bands for Synapsin, Syntaxin, and Draper are concerning and suggest the sensitivity of this method is low. To convince me that there is global signal over background, I would like to see MS analysis of samples in Figure 4B,D. For example, MS of streptavidin-pulldown from the following brain genotypes (1) elav>PGA + PhAc-OPP, (2) repo>PGA + PhAc-OPP, (3) elav>GFP + PhAc-OPP. Using quantitative approaches such as TMT labeling or SAINT analysis, I would expect to see enrichment of Draper to glia and Syntaxin to neurons, for example.

Considering our new data from analysis of total biotin-tagged neuronal/glial protein (new Figures 4B and 4E) alongside confocal imaging data from brains (Figure 2) and imaging data from other fly tissues such as muscle, trachea, salivary glands, fat body (Figure 3), we have now demonstrated that our method achieves robust signal over background for labeling newly-synthesized protein using two independent approaches. We have also shown that our method can be used to enrich cell typespecific labeled proteins from neurons and glia (now Figures 4C and 4F), thus verifying its use for assessing nascent proteomes of targeted cell populations. There are minor background bands for some (not all) enriched proteins, but these individual protein blots only detect near full-length labeled protein, whereas the total biotin blots should be able to detect all labeled protein or protein fragments and show a much greater signal-to-background strength (new Figures 4B and 4E). In future studies, we plan to use MS to query cell type-specific proteome changes under altered physiological or pathological conditions, yet it will require significant time, effort and funds to optimize and carry out MS studies. We believe that through the rigorous quality control experiments already performed, we have been able to effectively demonstrate high signal to background using two separate approaches, and therefore MS is not necessary for this purpose.

Reviewer #2 (Recommendations for the authors):I'm generally satisfied with the responses and changes to the manuscript. My only remaining request is that the authors include a line for PhAc-OPP in the Key Resources Table. Clearly, this is a key reagent, and the authors should be transparent that it is something that needs to be synthesized, which will likely limit how widespread the method is used by the community.

We are pleased that the changes made are satisfactory to the Reviewer. We have added PhAc-OPP to the Key Resources Table and cited the article in which its chemical synthesis is described in detail.